# Discovering Data Structures:
# Nearest Neighbor Search and Beyond

**Omar Salemohamed** *
Université de Montréal, Mila

**Laurent Charlin** †
HEC Montréal, Mila

**Shivam Garg**†
Microsoft Research

**Vatsal Sharan**†
University of Southern California

**Gregory Valiant**†
Stanford University

## Abstract

We explore if it is possible to learn data structures end-to-end with neural networks, with a focus on the problem of nearest-neighbor (NN) search. To answer this question, we introduce a framework for data structure discovery which adapts to the underlying data distribution and provides fine-grained control over query and space complexity. Crucially, the data structure is learned from scratch, and does not require careful initialization or seeding with candidate data structures. In several settings, we are able to reverse-engineer the learned data structures and query algorithms. For 1D nearest neighbor search, the model discovers optimal distribution (in)dependent algorithms such as binary search and variants of interpolation search. In higher dimensions, the model learns solutions that resemble k-d trees in some regimes, while in others, elements of locality-sensitive hashing emerge. Additionally, the model learns useful representations of high-dimensional data such as images and exploits them to design effective data structures. Beyond NN search, we believe the framework could be a powerful tool for data structure discovery for other problems, and adapt it to the problem of estimating frequencies over a data stream. To encourage future work in this direction, we conclude with a discussion on some of the opportunities and remaining challenges of learning data structures end-to-end.[3]

## 1 Introduction

*Can neural networks discover data structures from scratch?*

There are several motivations for this question. The first is scientific. Deep learning models are increasingly performing tasks once considered exclusive to humans, from image recognition and mastering the game of Go to engaging in natural language conversations. Designing data structures and algorithms, along with solving complex math problems, are particularly challenging tasks. They require searching through a vast combinatorial space with a difficult to define structure. It is therefore natural to ask what it would take for deep learning models to solve such problems. There are already promising signs: these models have discovered fast matrix-multiplication algorithms [1], solved SAT problems [2], and learned optimization algorithms for various learning tasks [3, 4, 5].

The second motivation is practical. Data structures are ubiquitous objects that enable efficient querying. Traditionally, they have been designed to be worst-case optimal and therefore agnostic to the underlying data and query distributions. However, in many applications there are patterns in these distributions that can be exploited to design more efficient data structures. This has motivated recent

---

*Corresponding Author: omar.salemohamed@gmail.com

†Authors listed in alphabetical order.

[3]Our code is available at: `https://github.com/omar-s1/data_structure_discovery`

39th Conference on Neural Information Processing Systems (NeurIPS 2025).

work on learning-augmented data structures which leverages knowledge of the data distribution to modify existing data structures with predictions [6, 7, 8, 9]. In much of this work, the goal of the learning algorithm is to learn distributional properties of the data, while the underlying query algorithm/data structure is hand-designed. Though this line of work clearly demonstrates the potential of leveraging distributional information, it still relies on expert knowledge to incorporate learning into such structures. It is natural to ask if we can go a step further and let deep learning models discover entire data structures and query algorithms in an end-to-end manner.

In this work, we investigate the possibility of end-to-end data structure discovery, with a focus on the nearest neighbor search problem. *A priori*, it is not obvious how to frame data structure discovery as an end-to-end learning problem. What are the underlying principles that unify different data structure problems? How can a data structure be represented and queried using neural networks? How do we ensure some notion of efficiency is enforced? Thus a main contribution of our work is in proposing how to frame the problem. To help chart the landscape of this paradigm and explore its scope and limits, we focus more on understanding the kind of algorithms that end-to-end learning discovers than results on any given benchmark.

## 1.1 Framework for data structure discovery

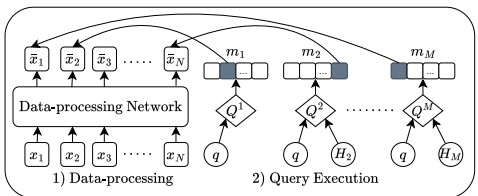

Figure 1: Our model has two components: **1)** A data-processing network transforms raw data into structured data, arranging it for efficient querying and generating additional statistics when given extra space (not shown in the figure). **2)** A query-execution network performs $M$ lookups into the output of the data-processing network to retrieve the answer to some query $q$. Each lookup $i$ is managed by a separate query model $Q^i$, which takes $q$ and the lookup history $H_i$, and outputs a one-hot lookup vector $m_i$ indicating the position to query.

Data structure problems are often divided into two steps: 1) construction and 2) query execution. The first step transforms a raw dataset $D$ into a structured database $\hat{D}$, while the second performs lookups into $\hat{D}$ to answer a query $q$. The performance of a data structure is typically quantified in terms of two measures: *space complexity*—the memory required to store the structure, and *query complexity*—the number of lookups needed to answer a query. One can typically tradeoff larger space complexity for smaller query complexity, and vice versa. We focus on these criteria as they are well-studied and directly impact practical efficiency.

To learn data structures, we have a *data-processing network* that learns how to map a raw dataset to a data structure, and a *query network* that learns an algorithm to answer queries using the data structure (Fig. 1). To ensure efficiency, we impose constraints on the data structure's size and the number of lookups the query network makes. Crucially, we propose end-to-end training of both networks such that the learned data structure and query algorithm are optimized for one another. In settings where it is beneficial to learn lower-dimensional representations from high-dimensional data, end-to-end training also encourages the representations to capture features that the data structure can exploit.

Why should learning data-structures end-to-end be possible? On one hand, jointly learning the data-processing and query networks end-to-end seems obvious, given the many successes of end-to-end learning over the past decade. On the other hand, it can be hard to imagine such learning getting off the ground. For instance, if the data-processing network produces a random garbled function of the dataset, the query model cannot do anything meaningful. In fact, we do observe cases where end-to-end learning fails. This challenge is further compounded by the discrete and combinatorial nature of how the query model accesses the data structure and the minimal supervision provided to the models. Despite these challenges, we find that in several settings end-to-end learning can recover a variety of classical algorithms.

## 1.2 Summary of Results

We focus our investigation on the problem of nearest neighbor (NN) search in both low and high dimensions. NN search is an ideal starting point for understanding the landscape of end-to-end data structure discovery given the extensive theoretical work on this topic, along with its widespread practical applications (e.g., similarity search over language/image embeddings [10]). Moreover, the

NN search problem has a rich design space and remains an active area of research [11, 12, 13]. In addition to NN search, we also explore the problem of frequency estimation in streaming data and discuss other potential applications for end-to-end data structure learning (Section 3).

Our findings are:

**Learning Classical Algorithms (Sections 2.2, 2.3)** For 1D nearest neighbor search, the data-processing network learns to sort, while the query network simultaneously learns to search over the sorted data. When the data follows a uniform or Zipfian distribution, the query network exploits this structure to outperform binary search. On harder distributions lacking structure, the network adapts by discovering binary search, which is worst-case optimal. Importantly, the model discovers that sorting followed by the appropriate search algorithm is effective for NN search in 1D without explicit supervision for these primitives. Similarly, in 2D, the model learns a data structure that outperforms k-d trees for uniform distributions and demonstrates recursive partitioning behavior, resembling k-d trees on harder distributions, by constructing medians along alternating dimensions.

**Useful representations in high dimensions (Section 2.4)** For high-dimensional data, the model learns representations that improve NN search efficiency. For instance, with data from a uniform distribution on a 30-dimensional hypersphere, it partitions space by projecting onto a pair of vectors, similar to locality-sensitive hashing. When trained on an extended 3-digit MNIST dataset, the model identifies features that capture number order, sorts images accordingly, and searches within the sorted set—all learned jointly from scratch! Moreover, it performs competitively with existing data structures on datasets like FashionMNIST and SIFT.

**Beyond NN search (Section 3)** To demonstrate broader applicability of the end-to-end learning paradigm, we study frequency estimation, a classic problem where a memory-constrained model observes a stream and estimates a query item's frequency. The learned structure leverages the data distribution to outperform baselines like CountMinSketch. Moreover, we use insights from the learned model to improve CountMinSketch on a practical dataset. We also outline several other problems for future work.

In summary, we take a first step toward end-to-end data structure learning, showing that it is not only feasible, but also capable of exploring a rich algorithmic space and recovering meaningful— sometimes even classical—solutions from scratch.

At the same time, our work has important limitations. Our experiments are limited to small-scale settings, and further work is needed to scale them up. Also, while we focus on enforcing query and space complexity constraints—arguably the most classical measures of efficiency—other aspects, such as preprocessing time and the inference cost of neural networks require more investigation. We discuss these limitations and potential solutions in Section 5.

## 2  Nearest Neighbor Search

Given a dataset $D = \{x_1, ..., x_N\}$ of $N$ points where $x_i \in \mathbb{R}^d$ and a query $q \in \mathbb{R}^d$, the nearest neighbor $y$ of $q$ is defined as $y = \arg\min_{x_i \in D} \ dist(x_i, q)$. We mostly focus on the case where $dist(\cdot)$ corresponds to Euclidean distance. Our objective is to learn a data structure $\hat{D}$ for $D$ such that given $q$ and a budget of $M$ lookups, we can output a (approximate) nearest neighbor of $q$ by querying at most $M$ elements in $\hat{D}$. When $M \geq N$, $y$ can be trivially recovered via linear search so $\hat{D} = D$ is sufficient. Instead, we are interested in the case when $M \ll N$.[4]

### 2.1  Setup

**Data-processing Network**   Recall that the role of the data-processing network is to transform a raw dataset into a data structure. The backbone of our data-processing network is an 8-layer transformer model based on the NanoGPT architecture [14] (see App B.1 for all architecture details). We use a quadratic-attention transformer to keep the data-processing model relatively general, however, we find that in certain settings it can be possible to use a cheaper alternative such as a transformer with linear attention [15]. See App. D.1 for a more detailed discussion.

---

[4]E.g. in 1D, binary search requires  $M = \log(N)$ lookups.

In the case of NN search, we want the data structure to preserve the original inputs and just reorder them appropriately as the answer to the nearest neighbor query should be one of elements in the dataset. The model achieves this by outputting a rank associated with each element in the dataset, which is then used to reorder the elements. More precisely, the transformer takes as input the dataset $D$ and outputs a scalar $o_i \in \mathbb{R}$ representing the rank for each point $x_i \in D$. These rankings $\{o_1, ..., o_N\}$ are then sorted using a differentiable sort function, $sort(\{o_1, o_2 ..., o_N\})$ [16, 17, 18], which produces a permutation matrix $P$ that encodes the order based on the rankings. By applying $P$ to the input dataset $D$, we obtain $\hat{D}_P$, where the input data points are arranged in order of their rankings. By learning to rank rather than directly outputting the transformed dataset, the transformer avoids the need to reproduce the exact inputs. Note that this division into a ranking model followed by sorting is without loss of generality as the overall model can represent any arbitrary ordering of the inputs. See App. E.2 for more information on why we learn permutations.

**Query Execution Network**   The role of the query-execution network is to output a nearest-neighbor of a query $q$ given a budget of $M$ lookups into the data structure $\hat{D}$. This introduces a combinatorial constraint—only a fixed number of discrete memory accesses are allowed—while we also require differentiability for gradient-based training, creating a tension between discreteness and optimization.

To implement this, the network consists of $M$ MLP query models $Q^1, ..., Q^M$. The query models do not share weights to keep the model relatively general, but we explore shared weights in App D.1. Each query model $Q^i$ outputs a one-hot vector $m_i \in \mathbb{R}^N$ which represents a lookup position in $\hat{D}$. To execute the lookup, we compute the value $v_i$ at the position denoted by $m_i$ in $\hat{D}$ as $v_i = m_i^\top \hat{D}$. In addition to the query $q$, each query model $Q^i$ also takes as input the query execution history $H_i = \{(m_1, v_1), ..., (m_{i-1}, v_{i-1})\}$ where $H_1 = \emptyset$. The final answer of the network for the nearest-neighbor query is given by $\hat{y} = m_M^\top \hat{D}$. To enforce exactly $M$ lookups while maintaining differentiability, we train with softmax-based soft lookups but add noise to their logits. We find this noise encourages sparser logits during training. At inference, we replace softmax with hardmax to produce exact one-hot vectors. In App. B.3 we provide some intuition for why we think adding noise to the logits induces sparsity.

**Data Generation and Training**   Each training example is a tuple $(D, q, y)$ consisting of a dataset $D$, query $q$, and nearest neighbor $y$ generated as follows: (i) sample dataset $D = \{x_1, ..., x_N\}$ from dataset distribution $P_D$, (ii) sample query $q$ from query distribution $P_q$, (iii) compute nearest neighbor $y = \arg\min_{x_i \in D} dist(x_i - q)$. Unless otherwise specified, $dist$ corresponds to the Euclidean distance. The dataset and query distributions $P_D, P_q$ vary across the different settings we consider and are defined later. Given a training example $(D, q, y)$, the data-processing network transforms $D$ into the data structure $\hat{D}$. Subsequently, the query-execution network, conditioned on $q$, queries the data structure to output $\hat{y}$. We use SGD to minimize either the squared loss between $y$ and $\hat{y}$, or the cross-entropy loss between the corresponding vectors encoding their positions. This is an empirical choice, and in some settings one loss function performs better than the other. All models are trained for at most 2 million gradient steps with early-stopping using a batch size of 1024. After training, we test our model on 10k inputs $(D, q, y)$ generated in the same way. See App B.1 for more details.

**Evaluation and Baselines**   We evaluate our end-to-end model (referred to as *E2E*) on 1-dimensional, 2-dimensional, and high-dimensional NN search. We primarily focus on data structures that do not use extra space, but we also explore scenarios with additional space in App B.12.

We compare against suitable NN data structures in each setting (e.g., sorting followed by binary search in 1D), and also against several ablations to study the impact of various model components. The *E2E (frozen)* model does not train the data-processing network, relying on rankings generated by the initial weights. The *E2E (no-permute)* model removes the permutation component of the data-processing network so that the transformer has to learn to transform the data points directly. The *E2E (non-adaptive)* model conditions each query model $Q^i$ on only the query $q$ and not the query history $H_i$. We select the prediction that is closest to the query as the final prediction $\hat{y}$.

## 2.2   One-dimensional data

**Uniform Distribution**   We consider a setting where the data distribution $P_D$ and query distribution $P_q$ correspond to the uniform distribution over $(-1, 1)$, $N = 100$ and $M = 7$. We plot the accuracy, which refers to zero-one loss in identifying the nearest neighbor, after each lookup in Fig. 2 (Left)

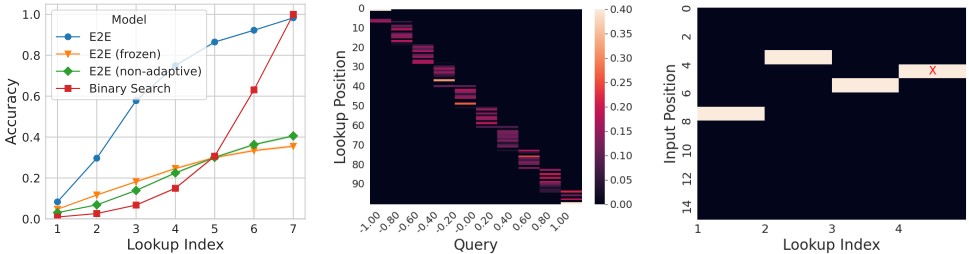

Figure 2: **(Left)** Our model (E2E) trained with 1D data from the uniform distribution over $(-1, 1)$ outperforms binary search and several ablations. **(Center)** Distribution of lookups by the first query model. Unlike binary search, the model does not always start in the middle but rather closer to the query's likely position in the sorted data. **(Right)** When trained on data from a "hard" distribution for which the query value does not reveal information about the query's relative position, the model finds a solution similar to binary search. The figure shows an example of the model performing binary search ('X' denotes the nearest neighbor location). See Fig. 13 for more examples.

(we include MSE plots as well in App. B.4). Recall that $v_i$ corresponds to the output of the $i$-th lookup. Let $v_i^*$ be the closest element to the query so far: $v_i^* = \arg\min_{v \in \{v_1, ..., v_i\}} ||v - q||_2^2$. For all the methods, we plot the nearest neighbor accuracy corresponding to $v_i^*$ for each lookup index $i$.

A key component of 1D NN search is sorting, and we observe that the trained model does indeed learn to sort effectively. We verify this by measuring the fraction of inputs mapped to the correct position in the sorted order, averaging 99.5% accuracy across multiple datasets. Remarkably, the model achieves this without explicit feedback to sort, learning the behavior through end-to-end training. While the separate sorting function aids this process, the model still has to learn to output the correct rankings.

The second key component is the ability to search over the sorted inputs. Here, our model learns a search algorithm that outperforms binary search, which is designed for the worst case. This is because unlike binary search, our model exploits knowledge of the data distribution to start its search closer to the nearest neighbor, similar to interpolation search [19]. For instance, if the query $q \approx 1$, the model begins its search near the end of the list (Fig. 2 (Center)). The minor sorting error ($\sim 0.5\%$) our model makes likely explains its worse performance on the final query.

To understand the relevance of different model components, we compare against various ablations. The *E2E (frozen)* model (untrained transformer) positions only about 9% of inputs correctly, explaining its under-performance. This shows that the transformer must learn to rank the inputs, and that merely using a separate function for sorting the transformer output is insufficient. The *E2E (non-adaptive)* baseline, lacking query history access, underperforms as it fails to learn adaptive solutions crucial for 1D NN search. The *E2E (no-permute)* ablation does not fully retain inputs and so we do not measure accuracy for this baseline. We verify this by measuring the average minimum distance between each of the transformer's inputs to its outputs. These ablations highlight the crucial role of both learned orderings and query adaptivity for our model.

**Hard Distribution** To verify that our model can also learn worst-case optimal algorithms such as binary search, we set $P_D$ to a "hard" distribution, where for any query, no strong prior exists over the position of its nearest neighbor in the sorted data (see App. B.6 for more details). To produce a problem instance, we first sample a dataset from $P_D$. We then generate the query by sampling a point (uniformly at random) from this dataset, and adding standard Gaussian noise to it. The hard distribution generates numbers at several scales, and this makes it challenging to train the model with larger $N$. Thus, we use $N = 15$ and $M = 3$. In general, we find that training models is easier when there is more structure in the distribution to be exploited.

The model does indeed discover a search algorithm similar to binary search. In Fig. 2 (Right), we show a representative example of the model's search behavior, resembling binary search (see Fig. 13 for more examples). The error curve in Fig. 11 also closely matches that of binary search.

Along with uniform and hard distributions, we show that end-to-end learning is possible also with a Zipfian distribution, which is common in many practical settings (see App. B.7).

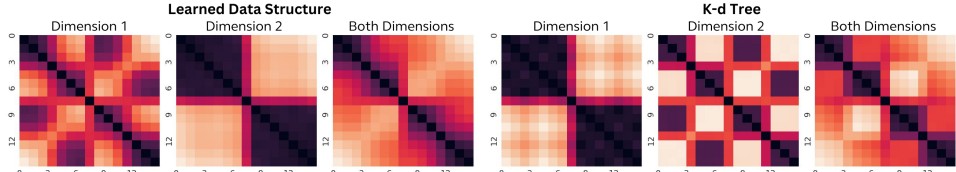

Figure 3: The learned data structure resembles a k-d tree in 2D. We show the average pairwise distances (along the first, second, and both dimensions) between points for the learned structure and the k-d tree, with darker colors indicating smaller distances. For the k-d tree, we arrange the points by in-order traversal. It recursively splits the points into two groups based on whether their value is smaller or larger than the median along a given dimension, alternating between dimensions at each level, starting with dimension 1. The learned data structure approximately mirrors this pattern, splitting by dimension 2 followed by dimension 1.

| Dataset | LSH | E2E | ITQ | KMeans | NeuralLSH |
|---|---|---|---|---|---|
| Hypersphere | 30.0 | 35.0 | 34.4 | 35.3 | 35.1 |
| FashionMNIST | 30.2 | 67.2 | 42.7 | 66.2 | 68.2 |
| SIFT | 14.3 | 46.9 | 30.1 | 47.1 | 46.7 |
| MNIST (3-digit) | 76.2 | 97.7 | - | - | - |

Table 1: NN search accuracy on high-d datasets compared to locality-sensitive hashing and learning-to-hash baselines. For a description of the baselines and a discussion on why we only compare with LSH on the MNIST dataset, see App. B.8

In summary, in all the above settings, starting from scratch, the data-processing network discovers that the optimal way to arrange the data is in sorted order. Simultaneously, the query-execution network learns to efficiently query this sorted data, leveraging the properties of the data distribution.

## 2.3 Two-dimensional data

Beyond one dimension it is less clear how to optimally represent a collection of points as there is no canonical notion of sorting along multiple dimensions. In fact, we observe in these experiments that different data/query distributions lead to altogether different data structures. This reinforces the value in learning both the data structure and query algorithm together, end-to-end.

**Uniform Distribution**    We use a setup similar to 1D, sampling both coordinates independently from the uniform distribution on $(-1, 1)$. We set $N = 100$ and $M = 6$, and compare to a k-d tree baseline. A k-d tree is a binary tree for organizing points in k-dimensional space, with each node splitting the space along one of the k axes, cycling through the axes at each tree level. Here, the E2E model achieves an accuracy of $75\%$ vs $52\%$ for the k-d tree (Fig. 7 in App. B.4). The model outperforms the k-d tree as it can exploit distributional information. By studying the permutations, we find that the model learns to put points that are close together in the 2D plane next to each other in the permuted order (see Fig. 9 for an example).

**Hard Distribution**    We also consider the case where both coordinates are sampled independently from the hard distribution used in the 1D setup (see Fig. 14 for the error curve). We observe that the data structure learned by the model is surprisingly similar to a k-d tree (see Fig 3). This is striking as a k-d tree is a non-trivial data structure, requiring recursively partitioning the data and finding the median along alternating dimensions at each level of the tree.

## 2.4 High-D data (Hypershere/FashionMNIST/SIFT)

High-dimensional NN search poses a challenge for traditional low-dimensional algorithms due to the curse of dimensionality. K-d trees, for instance, can require an exponential number of queries in high dimensions [20]. This has led to the development of approximate NN search methods such

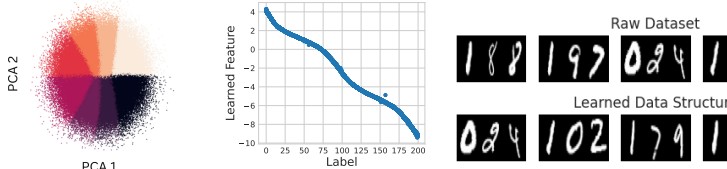

Figure 4: **(Left)** On the high-dimensional hypersphere, when trained with a single query, the model partitions the query space based on a projection onto two vectors, similar to LSH. We show the query projection onto the subspace spanned by these vectors and the lookup positions for different queries, where position is encoded by color. **(Center)** When trained end-to-end to do nearest neighbor search over 3-digit MNIST images, our model learns 1D features that capture the relative ordering of the numbers. **(Right)** Trained on 3-digit MNIST images, our data-processing model learns to sort the images without explicit supervision for sorting. While we train our model with datasets of size $N = 50$, we show a smaller instance with 5 images for better visualization.

as locality sensitive hashing (LSH) which have a milder dependence on dimension $d$ [21], relying on hash functions that map closer points in the space to the same hash bucket.

We train our model on datasets uniformly sampled from the $d$-dimensional unit hypersphere. The query is sampled to have a fixed inner-product $\rho \in [0, 1]$ with a dataset point. When $\rho = 1$, the query matches a data point, making regular hashing-based methods sufficient. For $\rho < 1$, LSH-based solutions are competitive. We train our model for $\rho = 0.8$ and compare it to LSH [22] and learning-to-hash baselines when $N = 100$, $M = 6$, and $d = 30$ (see App. B.8 for baseline details).

We observe that our model slightly outperforms the LSH baseline and performs competitively with the learning-to-hash baselines (Table 1). To better understand the data structure the model learns we consider a smaller setting where $N = 8$ and $M = 1$. We find that the model learns an LSH-like solution, partitioning the space by projecting onto two vectors in $\mathbb{R}^{30}$ (see Fig. 4 (Left)). Specifically, both the data-processing and query models learn the same partitioning of space in tandem. These partitions serve the same role as hash functions in LSH algorithms like SimHash [22]. Like LSH algorithms, to find a nearest-neighbor, the query model maps a point to its corresponding hash bucket and then compares it to other points in that bucket, ultimately selecting the closest one. We provide more details in App B.9.

We include additional high-dimensional (100D) experiments on two standard approximate NN benchmarks, FashionMNIST and SIFT [10], to demonstrate the model's performance on realistic data (see App B.10 for more details). Our model performs competitively with learning-to-hash baselines (Table 1), illustrating that E2E learning can recover reasonable solutions in a variety of settings, even when compared to carefully hand-designed solutions. Note that we do not expect E2E to surpass learning-to-hash baselines in these settings since query adaptivity does not appear to be helpful for these problems. Next, we consider a setting where query adaptivity is beneficial.

**Learning useful representations (MNIST)**  High-dimensional data often contains low-dimensional structure which can be leveraged to improve the efficiency of NN search. Here, we explore whether our end-to-end framework can learn representations that capture such structures, a challenging task that requires jointly optimizing the learned representation, data structure, and query algorithm.

We consider the following task: given a dataset of distinct 3-digit MNIST images (Fig.20) and a query image, find its nearest neighbor—defined as the image encoding the closest number to the query (i.e., nearest in label space). Instead of operating directly on pixel space, the data and query networks use low-dimensional representations learned by a CNN feature model $F$ (see App.B.11 for details).

Ideally, the feature model $F$ should learn 1d features encoding numerical order, the data model sorts them, and the query model performs a form of interpolation search using the fact that the data distribution is uniform to outperform binary search. This is almost exactly what is learned end-to-end, from scratch, without any explicit supervision about which image encodes which number. In Fig. 4 (Center) we plot the learned features. We find that the data model learns to sort the features (Fig. 4 (Right)) with 98% accuracy and the query model finds the nearest neighbor with $\approx 98\%$ accuracy (Table 1). Moreover, E2E learning beats LSH baselines as query adaptivity is useful here.

Notably, unlike hand-designed data structures that typically assume access to standard distance metrics (e.g., Euclidean), this approach requires no supervision on the underlying metric structure. The model only needs supervision on which dataset element is the nearest neighbor of a query, making it applicable even when the distance metric is unknown or implicit.

**Leveraging Extra Space**   Beyond learning effective orderings for querying, our framework can also learn to use extra memory to accelerate search. In App. B.12 we show that in both low and high-dimensions the data model can learn to pre-compute and store helpful statistics in extra space that enables the query model to find the nearest neighbor with less queries.

## 3   Beyond Nearest Neighbor Search

Next, we illustrate the broader applicability of the end-to-end learning paradigm by applying it to the classical problem of *frequency estimation in the streaming setting*. We then describe several other problems in Section 3.2 that this paradigm can be applied to.

### 3.1   Frequency Estimation

Given a stream of $T$ elements $e^{(1)}, ..., e^{(T)}$ from a universe, the task is to estimate the frequency of query element $e_q$ up to time-step $T$, aiming to minimize the mean absolute error between true and estimated counts. As in the NN setup, the key constraints are data structure size and the number of lookups for query execution, making this problem compatible with our framework. We explore frequency estimation as a second task mainly because it is structurally very different from NN search due to the streaming nature, yet E2E learning still works. Due to space constraints, we mainly discuss our findings here and leave implementation details to App C.2. We also discuss how both NN search and frequency estimation fit into our broader framework in App E.

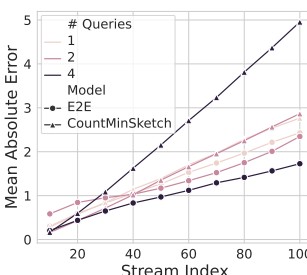

We evaluate our model in a setting where both stream and query distributions follow a Zipfian distribution, simulating frequency-estimation datasets where a few "heavy hitter" elements are queried more frequently [23]. For a given training sample, the rank order of elements is consistent across both stream and query distributions but randomized across different training samples. Consequently, the model cannot rely on specific elements being more frequent—only the overall Zipfian skew is consistent.

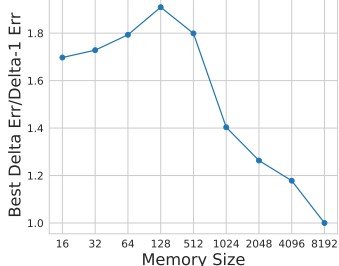

We compare our model with CountMinSketch (CMS), a hashing-based algorithm for frequency estimation [24] (See App. C.1 for an overview). Our model's performance improves with more queries and outperforms CMS (Figure 5).[5] We find that our model learns an algorithm similar to CMS but uses a smaller update delta,[6] which we hypothesize improves performance when collisions are frequent.

We use this insight to design a modified version of CMS that uses a custom update delta. In Fig. 5, we show that this augmented CMS algorithm can outperform vanilla CMS by up to a factor of $\approx 1.9\times$ on a real IP traffic dataset [25] consisting of a stream of 30 million IP addresses. These results demonstrate learning data structures end-to-end can provide useful insight into data structure design that can be transferred to realistic settings (see App C.3 for more details and App C.4 for additional experiments on frequency estimation in high-dimensional settings).

Figure 5: (Top) When estimating frequencies of elements drawn from a randomly ordered Zipfian distribution, our model outperforms the CMS baseline given 1, 2, and 4 queries. (Bottom) By augmenting CMS with an update delta ($\Delta$) $<$ 1 we can outperform vanilla CMS on the CAIDA IP dataset. For each memory size, we plot the relative performance of the best $\Delta$ vs the default, $\Delta = 1$.

---

[5]CMS degrades with more queries as for a fixed size memory ($k = 32$), it is more effective for this distribution to apply a single hash function over the whole memory than to split the memory into $k$ partitions of size $k/M$ and use separate hash functions.

[6]The update delta is the scalar increment value used to update the count of an item in the CMS hash table. For CMS, this value is 1. See C.3 for an overview of the CMS algorithm.

## 3.2 Other potential applications

Our framework is designed for problems that require efficiently answering queries over some collection of data. We believe it can be applied to any problem that shares this structure—namely, one where there is a data-processing step, a query algorithm, and constraints on space and query-time complexity. Here, we outline several such candidate problems that could benefit from this approach. We provide a more detailed discussion on the broader applicability of the framework and how it can be adapted to these problems in App. E.

**Graph data structures** Efficient graph representations are essential for connectivity or distance queries. Storing all distances between vertices in quadratic space enables $O(1)$ queries, but requires substantial memory. An alternative is storing the entire graph and running a shortest-path algorithm at query time. The challenge is finding a balance: using sub-quadratic space while still answering queries faster than full shortest-path computations [26].

**Sparse matrices** Another problem that can be framed as a data structure problem is compressing sparse matrices. Given an $M \times N$ matrix, on one hand, one can store the full matrix and access elements in $O(1)$ time. However, depending on the number and distribution of 0s in the matrix, data structures that use less than $O(MN)$ space could be designed. There is an inherent trade-off between how compressed the representation is and the time required to access elements of the matrix to solve various linear algebraic tasks involving the matrix, such as matrix-vector multiplication [27, 28].

**Learning statistical models** Our framework can handle problems such as learning statistical models, where the input to the data-processing network is a training dataset, and the output is a model such as a decision tree. The query algorithm would then access a subset of the model at inference time, such as by doing a traversal on the nodes of the decision tree. This could be used to explore optimal algorithms and heuristics for learning decision trees, which are not properly understood [29, 30].

## 4 Related Work

**Learning-Augmented Algorithms** Recent work has shown that traditional data structures and algorithms can be made more efficient by learning properties of the underlying data distribution. [31] introduced the concept of learned index structures which use ML models to replace traditional index structures in databases, resulting in significant performance improvements for certain query workloads. By learning the cumulative distribution function of the data distribution the model has a stronger prior over where to start the search for a record, which can lead to provable improvements to the query time over non-learned structures [32]. Other works augment the data structure with predictions instead of the query algorithm. For example, [8] use learned frequency estimation oracles to estimate the priority in which elements should be stored in a treap. Perhaps more relevant to the theme of our work is [33], which trains neural networks to learn a partitioning of the space for efficient nearest neighbor search using locality sensitive hashing, and the body of work on learned hash functions [34, 35, 11]. There is also work on learning the parameters of nearest neighbor search algorithms using machine learning [36] as well as works that have explored learning-augmented algorithms for frequency estimation [23, 37]. While these works focus on augmenting data structure design with learning, we explore whether data structures can be discovered entirely end-to-end using deep learning. Our approach eliminates the human-in-the-loop, making it promising for settings with limited insights into suitable data structures. However, this comes at the cost of losing the provable guarantees that learning-augmented methods typically offer.

**Neural Algorithmic Learners** There is a significant body of work on encoding and learning algorithms with neural networks. Graves et al. [38] introduced the Neural Turing Machine which uses external memory to learn tasks such as sorting. The neural algorithmic reasoning line of work [39, 40, 41, 42] aims to simulate a variety of classical algorithms using neural networks. These works typically train models with intermediate outputs of the algorithm they are trying to learn whereas we rely on minimal supervision. There has also been work on learning algorithms in an end-to-end fashion. Fawzi et al. [1] train a model using reinforcement learning to discover matrix multiplication algorithms, while Selsam et al. [2] train neural networks to solve SAT problems. Garg et al. [3] show that transformers can be trained to encode learning algorithms for function classes such as linear functions and decision trees. A recent line of work on neural sketching algorithms [43, 44, 45] shares several similarities to our frequency estimation experiments. Both these methods and ours can be viewed as memory-augmented neural networks that learn to read and write from a differentiable

memory. This line of work is more focused on developing scalable sketching algorithms whereas we explore whether or not neural networks can discover such algorithms from scratch with minimal inductive biases. See App C.6 for additional discussion.

# 5 Discussion

**Limitations**    While our work takes a first step in exploring end-to-end data structure discovery, there are several important limitations.

*Efficiency:* Our investigation focuses on the space and query efficiency of the learned data structure. However, in practice, factors such as the neural net inference cost and the pre-processing time to construct the data structure are also important. For example, using standard Transformers in the data model leads to pre-processing time that scales quadratically with the number of items. In preliminary experiments, we find that substituting the quadratic attention transformer with a more efficient alternative, such as linear attention (App. D.1), is often feasible. Further exploration of these architectural choices is an important direction for future work. We also benchmark the computational overhead of our methods in App D.3.

*Scale:* Due to computational constraints, most of our experiments are conducted on datasets with $N = 100$ (with some scaled to $N = 500$; see App. D.2). Practical end-to-end deployment would require further scaling, which remains an open challenge. While increasing both the model sizes and training time will help to scale to larger datasets to some extent, it will also be important to explore new ideas as well such as better inductive biases that enable smaller models to handle large datasets more effectively. For instance, sharing weights among query models may help scale the number of queries. See App. D.2 for a more detailed discussion on scaling and preliminary experiments.

In addition to the above limitations, learned data structures lack the same level of interpretability and provable guarantees as their classical counterparts. However, in practical settings, the benefits of learned distribution-dependent algorithms may outweigh these limitations.

**Future Applications**    While scaling is necessary for practical deployment, small-scale models can still yield valuable insights that generalize to larger settings. For example, an *AI-in-the-loop* discovery workflow might: (1) train a model on a target problem to surface promising heuristics (e.g., space partitioning criteria), (2) analyze its behavior to extract actionable insights, and (3) validate those insights theoretically or empirically. Such workflows are feasible even with small-scale models. In Section 3, for instance, we use insights from our model to design an augmented version of CMS that outperforms the original algorithm on a practical dataset. Moreover, the ability of E2E learning to recover classical algorithms such as binary search, k-d trees, and LSH shows that these models can effectively explore the data structure design space, highlighting the promise of this approach.

**Conclusion**    We set out by asking: can neural networks discover data structures? At the outset, many basic questions were unclear. How should data structures and queries be represented using neural networks—especially under combinatorial constraints like limited query complexity? Even with a reasonable setup, it was not clear if training would take off, as both networks are trained from scratch and depend on each other for useful signal. And even if learning does take off, can it explore the rich space of possible data structures and algorithms?

To our surprise, once we set up the training with space/query constraints and a structured lookup interface the model did learn meaningful algorithms. It recovered classic data structures like sorting, k-d trees, and LSH-like behavior, often adapting them to the data distribution. In more complex cases like the extended MNIST experiment, it learned image features, built a data structure over them, and learned a query network to search it—all from scratch and with minimal supervision.

We view this as the main contribution of the paper: framing data structure discovery as an end-to-end learning problem and showing that it not only works, but can explore a rich solution space. We hope this research encourages further exploration of end-to-end learning for data structures—from tackling efficiency and scaling challenges to applying the framework to new problems and leveraging it as an *AI-in-the-loop* tool for discovery.

## Acknowledgments

OS and LC acknowledge the support of the CIFAR AI Chair program. This research was also enabled in part by compute resources provided by Mila – Quebec AI Institute (mila.quebec). GV is supported by a Simons Foundation Investigator Award, NSF award AF-2341890 and UT Austin's Foundations of ML NSF AI Institute. VS was supported by NSF CAREER Award CCF-2239265, an Amazon Research Award, a Google Research Scholar Award and an Okawa Foundation Research Grant. This work was done in part when VS was visiting the Simons Institute for the Theory of Computing.

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

# Appendix

# A  Related Work

**Learning-Augmented Algorithms**   Recent work has shown that traditional data structures and algorithms can be made more efficient by learning properties of the underlying data distribution. [31] introduced the concept of learned index structures which use ML models to replace traditional index structures in databases, resulting in significant performance improvements for certain query workloads. By learning the cumulative distribution function of the data distribution the model has a stronger prior over where to start the search for a record, which can lead to provable improvements to the query time over non-learned structures [32]. Other works augment the data structure with predictions instead of the query algorithm. For example, [8] use learned frequency estimation oracles to estimate the priority in which elements should be stored in a treap. Perhaps more relevant to the theme of our work is [33], which trains neural networks to learn a partitioning of the space for efficient nearest neighbor search using locality sensitive hashing, and the body of work on learned hash functions [34, 35, 11]. There is also work on learning the parameters of nearest neighbor search algorithms using machine learning [36]. There has also been a number of works that have explored learning-augmented algorithms for frequency estimation [23, 37]. While all these works focus on augmenting data structure design with learning, we explore whether data structures can be discovered entirely end-to-end using deep learning. Our approach eliminates the human-in-the-loop, making it promising for settings with limited insights into suitable data structures. However, this comes at the cost of losing the provable guarantees that learning-augmented methods typically offer.

**Neural Algorithmic Learners**   There is a significant body of work on encoding algorithms into deep networks. Graves et al. [38] introduced the Neural Turing Machine (NTM), which uses external memory to learn tasks like sorting and copying. Veličković et al. [39] used graph neural networks (GNNs) to encode classical algorithms such as breadth-first search. These works train deep networks with a great degree of supervision with the aim of encoding known algorithms. For instance, Graves et al. [38] use the ground truth sorted list as supervision to train the model to sort. There has also been work on learning algorithms in an end-to-end fashion. Fawzi et al. [1] train a model using reinforcement learning to discover matrix multiplication algorithms, while Selsam et al. [2] train neural networks to solve SAT problems. Garg et al. [3] show that transformers can be trained to encode learning algorithms for function classes such as linear functions and decision trees. Our work adds to this line of research on end-to-end learning, focusing on discovering data structures.

# B  Nearest Neighbor Experiments

## B.1  Training Details

The transformer in the data-processing network is based on the NanoGPT architecture [14] and has 8 layers with 8 heads each and an embedding size of 64. Instead of using discrete tokens, we have input and output projection layers (Linear layers) that project the input to the hidden dimension and from the hidden dimension to the scalar output values. Each query model $Q_\theta^i$ is a 3-layer MLP with a hidden dimension of size 1024. Each hidden layer consists of a linear mapping followed by LayerNorm [46] and the ReLU activation function [47]. In all experiments we use a batch size of 1024, 1e-3 weight decay and the Adam optimizer [48] with default PyTorch [49] settings. We do a grid search over $\{0.0001, 0.00001, 0.00005\}$ to find the best learning rate for both models. Near the end of training when performance plateaus, we decrease learning rates to boost performance. We apply the Gumbel Softmax [50] with a temperature of 2 to the lookup vectors to encourage sparsity. Apart from the learning rate of the two models, we did very minimal hyperparameter tuning as the majority of experiments worked without additional tuning. We expect additional performance gains would come with further tuning but are more interested in understanding the general performance landscape relative to algorithmic baselines rather than optimizing any single result. All experiments are run on a single NVIDIA RTX8000 GPU.

**Training Steps/Data set Sizes**   As we have direct access to the dataset distribution, for all experiments (except those with real high-dimensional data), we do not use a fixed training set but rather train 'online', sampling new (training) datasets directly from the dataset distribution. The number of optimization steps we use varies across experiments but is up to 500k for most experiments with $N \leq 100$, though for experiments with $N = 500$ we ran for up to 2 million steps (with early

stopping). For experiments with fixed training sets (e.g. 3-Digit MNIST/FashionMNIST/SIFT), we resampled from their corresponding training sets (sizes 60k/60k/1M, respectively) and used their test sets for evaluation. Given that the models perform well on test data for these distributions, it suggests that we do not need as much unique data as we use in the online setting. We leave an exploration for this to future work.

## B.2 Training Variance

We found that the setup is robust to random model seeds, however, below we include plots of 1D/2D/30D experiments using three different seeds to showcase the variation.

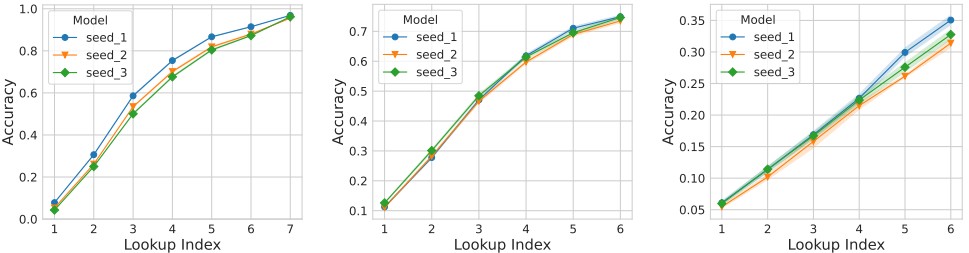

Figure 6: 1D (Uniform), 2D (Uniform), 30D (Hypersphere) experiments with different seeds.

## B.3 Noise Injection for Lookup Sparsity

We find that adding noise prior to applying the soft-max on the lookup vector $m_i$ leads to sparser queries. We hypothesize that this is because the noise injection forces the model to learn a noise-robust solution, and the softmax output becomes robust when the logits are well-separated. Well-separated logits, in turn, lead to sparser solutions.

Consider a simplified setup in 1D where the query model is not conditioned on $q$ and is only allowed one lookup ($M = 1$) and $D$ is a sorted list of three elements: $D = [x_1, x_2, x_3]$. For a given query $q$ and its nearest neighbor $y$, the query-execution network is trying to find the optimal vector $\hat{m} \in \mathbb{R}^3$ that minimizes $||y - m^T D||_2^2$ where $m = softmax(\hat{m} + \epsilon), \epsilon \sim$ Gumbel distribution [50]. Given that $M = 1$, the model cannot always make enough queries to identify $y$ and so in the absence of noise the model may try to predict the 'middle' element by setting $\hat{m}_1 = \hat{m}_2 = \hat{m}_3$. However, when noise is added to the logits $\hat{m}$ this solution is destabilized. Instead, in the presence of noise, the model can robustly select the middle element by making $\hat{m}_2$ much greater than $\hat{m}_1, \hat{m}_3$. We test this intuition by running this experiment for large values of $N$ and find that with noise the average gradient is much larger for $\hat{m}_{N/2}$.

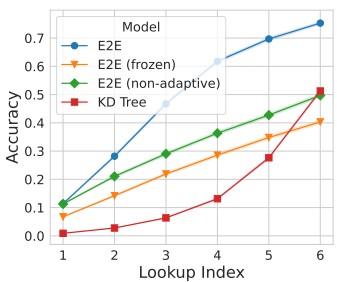

Figure 7: 2D Uniform Accuracy.

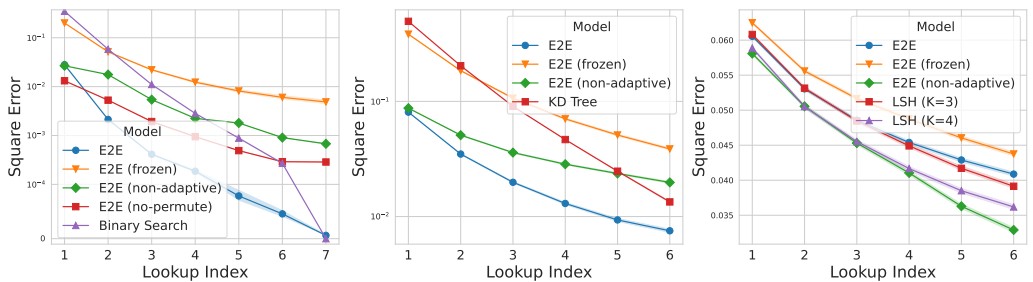

Figure 8: Mean square error plots for **(Left)** 1D Uniform distribution, **(Center)** 2D Uniform distribution, **(Right)** 30D Uniform distribution over unit hyper-sphere.

**B.5 2D Uniform Distribution**

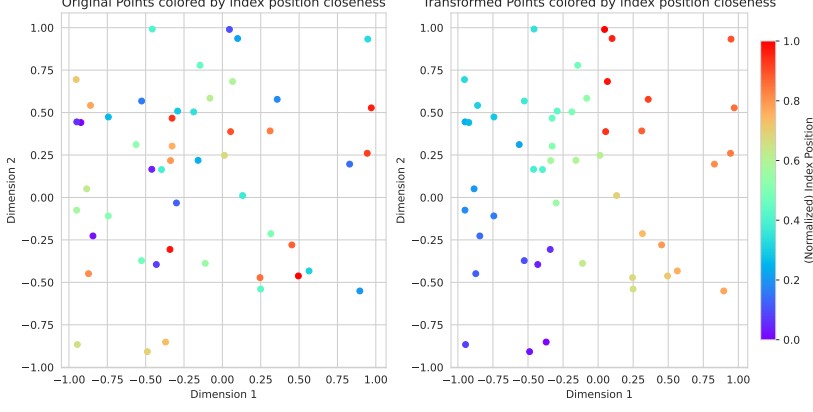

Figure 9: Our model's learned permutation on the 2D uniform distribution. The model puts elements that are close together in the Euclidean plane next to each other in the permutation.

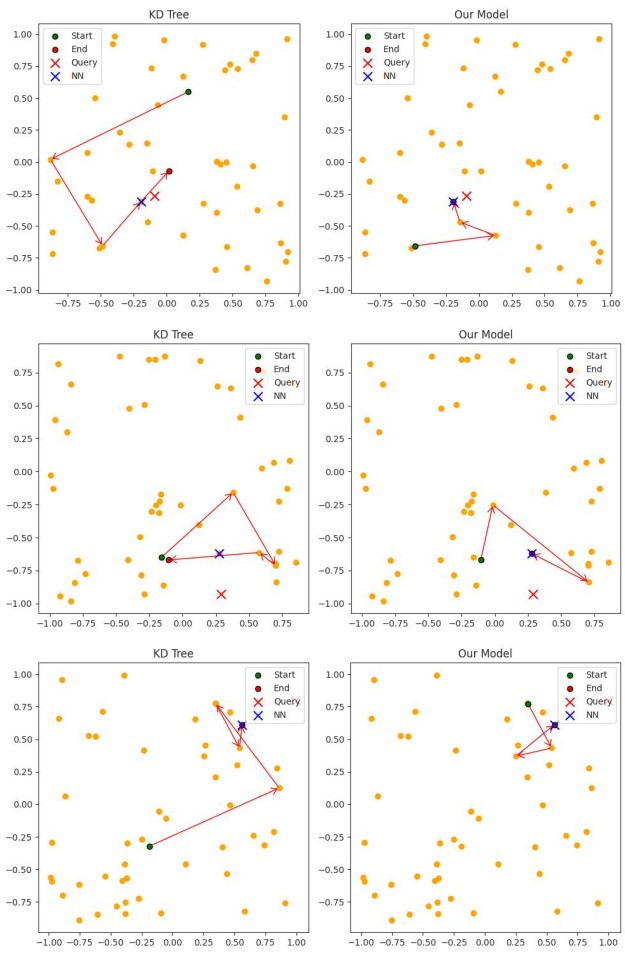

Figure 10: k-d search vs. our model on the uniform distribution in 2D. Unlike the k-d tree, our model has a stronger prior over where to begin its search.

## B.6 Hard Distribution

To generate data from the hard distribution, we first sample the element at the 50th percentile from the uniform distribution over a large range. We then sample the 25th and 75th percentile elements from a smaller range and so on. The intuition behind this distribution is to reduce concentration such that $p(NN|q)$ is roughly uniform where $NN$ denotes the index of the nearest-neighbor of $q$ in the sorted list.

Precisely, to sample $N$ points from the hard distribution we generate a random balanced binary tree of size $N$. All vertices are random variables of the form $Uniform(0, a^{\log n - k})$ where $a$ is some constant and $k$ is the level in the tree that the vertice belongs to. If the $i - th$ node in the tree is the left-child of its parent, we generate the point $x_i$ as $x_i = x_{p(i)} - d_i$ where $p(i)$ denotes the parent of the $i - th$ node and $d_i$ is a sample from node $i$ of the random binary tree. Similarly, if node $i$ is the right child of its parent, $x_i = x_{p(i)} + d_i$. For the root element $x_0 = d_0$. In our experiments we set $a = 7$. The larger the value of $a$, the greater the degree of anti-concentration. We found it challenging to train models with $N > 16$ as the range of values that $x_i$ can take increases with $N$. Thus for larger $N$, the model needs to deal with numbers at several scales, making learning challenging.

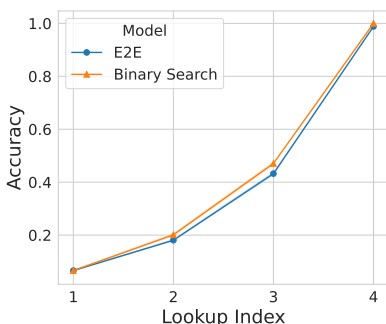

Figure 11: Our model's performance is closely aligned with binary search on the hard distribution in 1D. By design, this distribution does not have a useful prior our model can exploit and so it learns a binary search like solution.

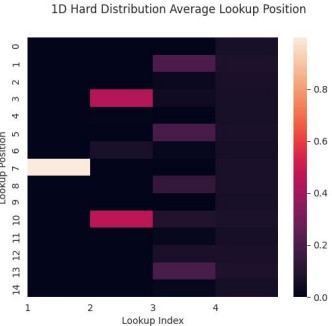

Figure 12: The positional distribution per lookup in the 1D Hard experiment. Our model closely aligns with binary search, first looking at the middle element, then (approximately) either the 25th or 75th percentile elements, and so on.

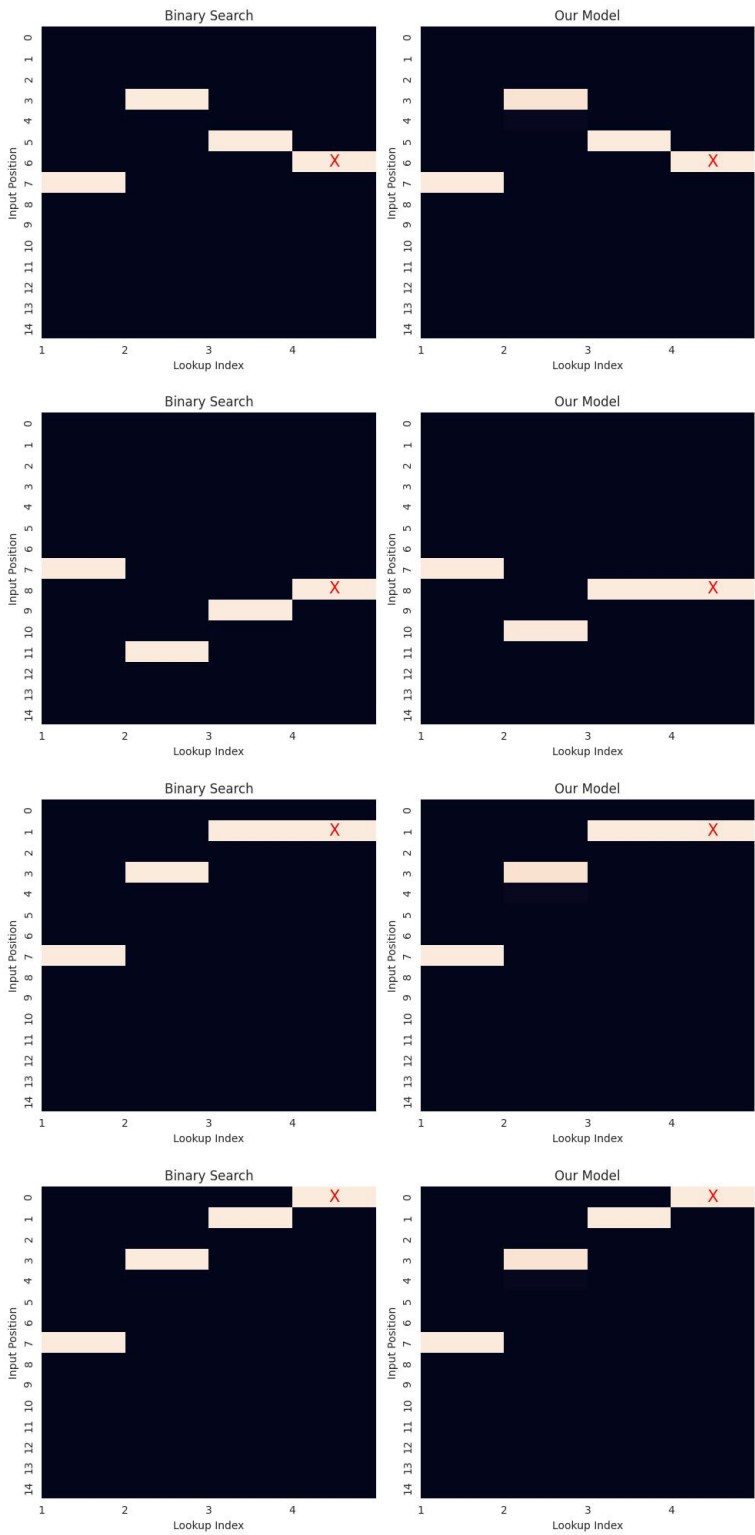

Figure 13: Binary Search vs. our model on the hard distribution in 1D.

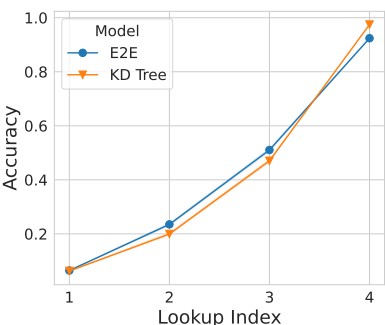

Figure 14: On the 2D hard distribution our model roughly tracks the performance of a k-d tree.

## B.7  1D Zipfian Experiment

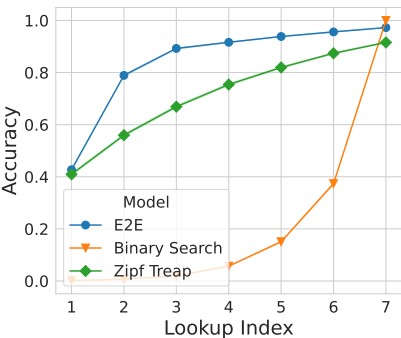

Figure 15: For 1D Zipfian query distribution, our model performs slightly better than the the learning-augmented treap algorithm from [13] and both methods significantly outperforms binary search.

Prior work has shown that several real-world query distributions follow a Zipfian trend whereby a few elements are queried far more frequently than others, leading to the development of learning-augmented algorithms aimed at exploiting this [13]. We consider a setting where $P_D$ is the discrete uniform distribution over $\{1, ..., 200\}$ and $P_q$ is a Zipfian distribution over $\{1, ..., 200\}$ skewed towards smaller numbers such that the number $i$ is sampled with probability proportional to $\frac{1}{i^\alpha}$. We set $\alpha = 1.2$. Again, in this setting $N = 100$ and $M = 7$.

In Figure 15 we compare our model to both binary search and the learning-augmented treap from Lin et al. [8]. Our model performs slightly better than the learning-augmented treap and both algorithms significantly outperform binary search with less than $\log(N)$ queries. This setting highlights a crucial difference in spirit between our work and much of the existing work on learning-augmented algorithms. While the Zipfian treap incorporates learning in the algorithm, the authors still had to figure out how an existing data structure could be modified to support learning. On the other hand, by learning end-to-end, our framework altogether removes the need for the human-in-the-loop. This is promising as it could be useful in settings where we lack insight on appropriate data structures. The flip side, however, is that learning-augmented data structures usually come with provable guarantees which are difficult to get when training models in an end-to-end fashion.

## B.8  Hashing Baselines

### B.8.1  Locality Sensitive Hashing (LSH)

Our LSH baseline is based on the SimHash algorithm by [22]. SimHash is particularly relevant for our hypersphere setting as the algorithm is designed for nearest-neighbor retrieval based on cosine similarity and all the points in our experiment fall on the unit hypersphere so minimizing euclidean distance is equivalent to minimizing the cosine similarity. We construct the LSH baseline as follows. We sample $K$ random vectors $\mathbf{r_1}, ..., \mathbf{r_K}$ from the standard normal distribution in $\mathbb{R}^d$. For a given vector $\mathbf{v} \in \mathbb{R}^d$, its hash code is computed as $hash(\mathbf{v}) = [sign(\mathbf{v^T r_1}), ..., sign(\mathbf{v^T r_K})]$. In total, there are $2^K$ possible hash codes. To create a hash table, we assign each hash code a bucket of size $N/2^K$. For a given dataset $D = \{x_1, ..., x_N\}$, we place each input in its corresponding bucket (determined by its hash code $hash(x_i)$. If the bucket is full, we place $x_i$ in a vacant bucket chosen at random. Given a query $q$ and a budget of $M$ lookups, the baseline retrieves the first $M$ vectors in the bucket corresponding to $hash(q)$. If there are less than $M$ vectors in the bucket, we choose the remaining vectors at random from other buckets. We design this setup like so to closely align with the constraints of our model (i.e. only learning a permutation). Note that for all experiments that compare to LSH, we choose the value of $K$ that maximizes the performance of LSH.

### B.8.2  Learning to Hash

In addition to locality-sensitive hashing, we include comparisons to several learning-to-hash baselines: ITQ, K-Means, and NeuralLSH. See Dong et al. [33] for more details about these specific baselines.

Similar to classical LSH methods, learning-to-hash methods aim to hash elements to buckets such that close by elements are more likely to be mapped to the same buckets. The main difference is that learning to hash methods learn the hash function from data while it is traditionally hand-designed. However, note that while our framework learns the whole data structure and an adaptive query algorithm in an end to end manner, the learning to hash methods just learn part of the data structure while the query algorithm is the same as that of LSH and thus is not adaptive.

### B.8.3 Baseline Choices for MNIST Experiment

For the 3-digit MNIST setting, we exclude the learning-to-hash methods as they cannot be applied directly. Specifically, in our MNIST setup the metric is not given but implicitly provided via nearest neighbor supervision. This means our model needs to learn the metric space, which learning-to-hash methods are not designed for. However, to highlight our models' strengths in this regime, we compare with an oracle hash function (denoted as LSH in the table) - splitting the 200 number universe evenly into K clusters where the first N/K elements are assigned to cluster 1, the second N/K elements to cluster 2 and so forth. Note that no learning-to-hash method could outperform this oracle hash function as it is an upper-bound with access to the correct metric. To execute a NN search, a query is mapped to its corresponding bucket and the element closest to this number (defined over this 1D label space) is chosen as the NN. When N=50 and M=7 we find that the optimal K is 8.

In Figure 19, we also compare with binary search directly operating on the digit space. To be clear, these are unfair comparisons as our model has to learn features that capture the underlying metric space but in this case it is provided directly for the two baselines. The oracle hashing baseline (and thus any learning-to-hash baseline) should still underperform our model in this regime as it does not use an adaptive query algorithm which is necessary for achieving the $O(logN)$ performance our model recovers.

### B.9 N=8, M=1 30D LSH Experiment

To determine if our model has learned an LSH-like solution, we try to reverse engineer the query model in a simple setting where $N = 8$ and $M = 1$. The query-execution model is only allowed one lookup. We fit 8 one-vs-rest logistic regression classifiers using queries sampled from the query distribution and the output of the query model (lookup position) as features and labels, respectively. We then do PCA on the set of 8 classifier coefficients. We find that the top 2 principal components explain all of the variance which suggests that the query model's mapping can be explained by the projection onto these two components. In Figure 16 we plot the projection of queries onto these components and color them based on the position they were assigned by the query model. We do the same for inputs $x_i \in D$ and color them by the position they were permuted to. The plot on the right suggests that the data-processing network permutes the input vectors based on their projection onto these two components. This assignment is noisy because there may be multiple inputs in a dataset that map to the same bucket and because the model can only store a permutation, some buckets experience overflow. Similarly, the query model does a lookup in the position that corresponds to the query vector's bucket. This behaviour suggests the model has learned a locality-sensitive hashing type solution! Specifically, both the data-processing and query models learn the same partitioning of space in tandem. These partitions serve the same role as hash functions in LSH algorithms like SimHash [22]. Like LSH algorithms, to find a nearest-neighbor, the query model maps a point to its corresponding hash bucket and then compares it to other points in that bucket, ultimately selecting the closest one.

### B.10 Additional High-Dimensional NN Experiments with Realistic Data

We run additional nearest-neighbor experiments on two standard high-dimensional approximate nearest-neighbor benchmarks: SIFT and FashionMNIST (projected to 100 dimensions via PCA) [10] to further demonstrate our model can handle more realistic data. The SIFT dataset contains high-dimensional feature descriptors extracted using the Scale-Invariant Feature Transform (SIFT) algorithm. It is commonly used for nearest neighbor search, image retrieval, and feature matching tasks. The FashionMNIST dataset consists of 28×28 grayscale images of clothing items, such as shoes, shirts, and dresses.

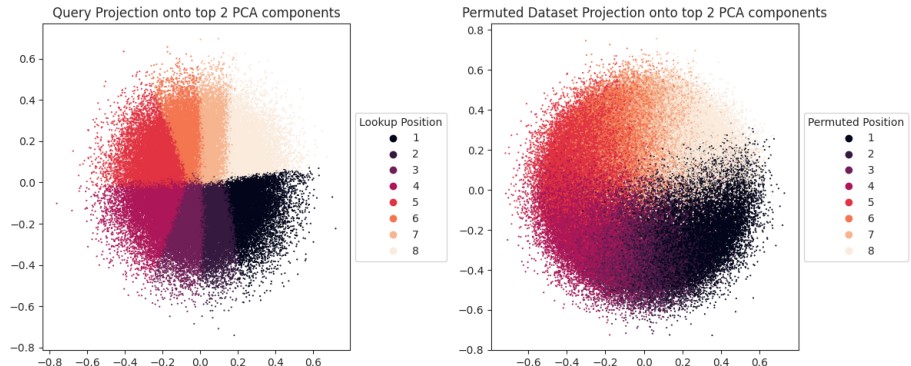

Figure 16: **(Left)** Projection of queries onto top two PCA components of the decision boundaries of the query model, colored by the lookup position the query is mapped to. **(Right)** Projection of inputs onto the same PCA components colored by the position the data-processing model places them in. Both the data-processing and query models map similar regions to the same positions, suggesting an LSH-like bucketing solution has been learned.

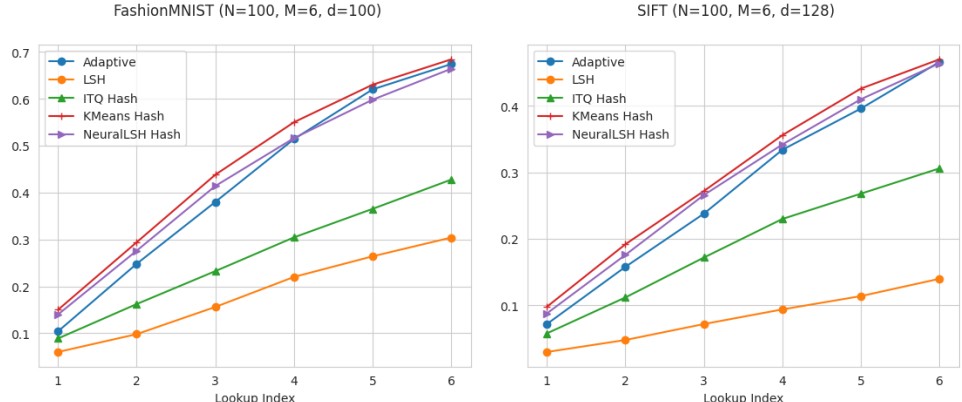

Figure 17: Our model compared to various hashing baselines on the FashionMNIST dataset (**left**) and the SIFT dataset (**right**). At the final query, our E2E solution performs as well as the hashing baselines. At intermediate lookups, it slightly under-performs, however, the model is only being trained to optimize performance at the final lookup. We do not expect our model to outperform these baselines in this setting as it is unclear that query adaptivity should be beneficial here. Rather, we include these results to further emphasize that our E2E model can recover reasonable solutions in a variety of settings - even when compared to carefully hand-designed solutions.

For each dataset, we use the train split to train our model (as well as the learning-to-hash baselines) and for evaluation we use the test split. The learning-to-hash baselines (described in B.8.2) cluster the data into 16 partitions and we then use the same setup described in Appendix B.8.1 to execute the NN search. We chose 16 partitions as this produces the best performance for the learning-to-hash baselines in our setup where $N = 100$ and $M = 6$.

Our model performs competitively with these learning-to-hash baselines (Figure 17). We do not expect our model to outperform these baselines in this setting as it is unclear that query adaptivity should be beneficial here. Rather, we include these results to further emphasize that end-to-end learning can recover reasonable solutions in a variety of settings - even when compared to carefully hand-designed solutions. We also show an example of how the data-processing model learns to transform the FashionMNIST data by organizing the images by class (Figure 18).

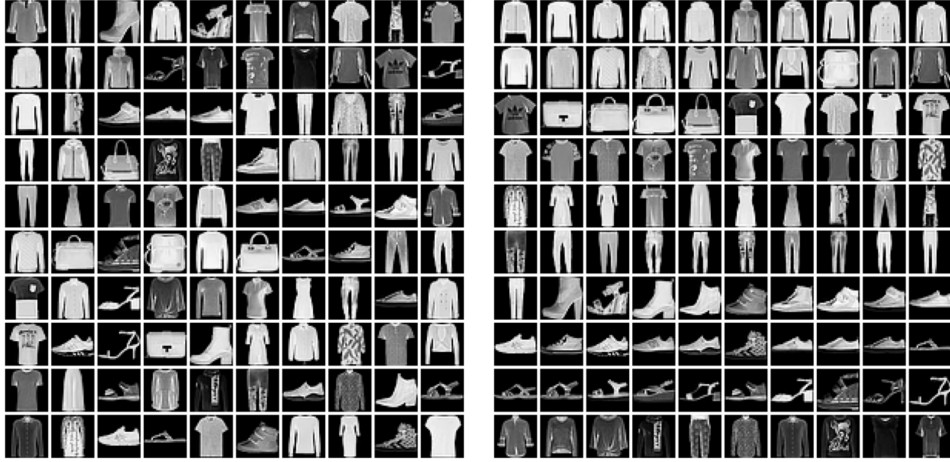

Figure 18: (**Left**) Sample raw dataset from FashionMNIST (**Right**) The learned data structure. The data-processing model learns to cluster similar items together.

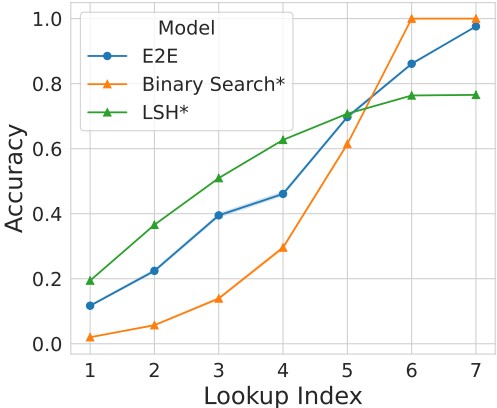

Figure 19: 3-Digit MNIST Nearest Neighbors Accuracy. Even though binary search (over the underlying digits) is an unfair comparison, we include it as a reference to compare our model's performance with.

### B.11 MNIST (3-digit) Experiments

We generate images of 3-digit numbers by concatenating digits from MNIST (see Fig. 20 for image samples). To construct a nearest-neighbor dataset $D$, we sample $N = 50$ labels (each label corresponds to a number) uniformly from 0 to 199. For each label, we then sample one of its associated training images from 3-digit MNIST. Additionally, we sample a query label (uniformly over $\{0, .., 199\}$) and its corresponding training image and find its nearest neighbor in $D$, which corresponds to the image with the closest label. We emphasize that the model has no label supervision but rather only has access to the query's nearest neighbor. After training, we evaluate the model using the same data generation process but with images sampled from the 3-digit MNIST test set.

As both the data-processing and query-execution networks should operate over the same low-dimensional representation we train a CNN feature model (architecture described below) $F_\phi$ as well. Our setup remains the same as before except now the data-processing network and query-execution network operate on $\{F_\phi(x_1), ..., F_\phi(x_N)\}$ and $F_\phi(q)$, respectively. As the underlying distance metric does not correspond to the Euclidean distance, we minimize the cross-entropy loss instead of the MSE loss. Note that the cross-entropy loss only requires supervision about the nearest

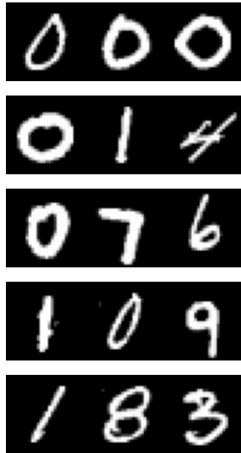

Figure 20: Samples from 3-Digit MNIST

| Model | 0 | 2 | 4 | 8 | 16 | 32 | 64 | 128 |
|---|---|---|---|---|---|---|---|---|
| Bucket Baseline | - | 5.9 | 11.7 | 23.6 | 44.5 | 65.8 | 81.1 | 89.5 |
| E2E | 44.0 | 47.9 | 52.0 | 58.8 | 71.9 | 75.3 | 88.4 | 91.3 |

Table 2: 1D NN search accuracy given varying amounts of extra space. The E2E model effectively uses extra space and outperforms a bucketing baseline.

neighbor of the query, and does not require the exact metric structure, so it can be used even where the exact metric structure is unknown.

**CNN Feature Model Architecture** $F_\phi$

```
Conv2d(in_channels=1, out_channels=32, kernel_size=3, stride=1, padding=1)
ReLU()
MaxPool2d(kernel_size=2, stride=2, padding=0)
Conv2d(in_channels=32, out_channels=64, kernel_size=3, stride=1, padding=1)
ReLU()
MaxPool2d(kernel_size=2, stride=2, padding=0)
Linear(in_features=64*7*21, out_features=128)
ReLU()
Linear(in_features=128, out_features=1)
```

We plot the results in Fig 19 compared to an optimal hashing and binary search baseline. Note these are not fair comparisons as these baselines operate over numbers directly instead of their corresponding images and are only provided for comparison. Please refer to B.8.3 for a more detailed discussion on this.

## B.12 Extra Space

We also consider scenarios where the data structure can use additional space. To support this use case, the data-processing transformer outputs $T$ extra vectors $b_1, ..., b_T \in \mathbb{R}^d$ which can be retrieved by the query-execution network in the same way as the other outputs. We form the data structure $\hat{D}$ by concatenating the permuted inputs and the extra vectors: $\hat{D} = [\hat{D}_P, b_1, ..., b_T]$.

### B.12.1 1D Extra Space

Most of our nearest neighbor experiments show our model's ability to learn useful orderings for efficient querying. Data structures, however, can also leverage pre-computed information to accelerate search. For example, with infinite space, a data structure could store the nearest neighbor for every

possible query, enabling $O(1)$ search. Here, we test whether the model can effectively use additional space.

We run an experiment where the data and query distribution are uniform over $(-1, 1)$ with $N = 50, M = 2$. We allow the data-processing network to output $T \in \{0, 2^1, 2^2, 2^3, 2^4, 2^5, 2^6, 2^7\}$ numbers $b_1, ..., b_T \in \mathbb{R}$ in addition to the $N$ rankings. We plot the NN accuracy as a function of $T$ in Table 2 compared to a simple bucketing baseline (explained in App B.12.2).

Our model's accuracy monotonically increases with extra space demonstrating that the data-processing network learns to pre-compute useful statistics that enable efficient querying. We provide some insights into the learned solution in App B.12.1 and show that our model can be trained to use extra space in the high-dimensional case as well (App B.12.4).

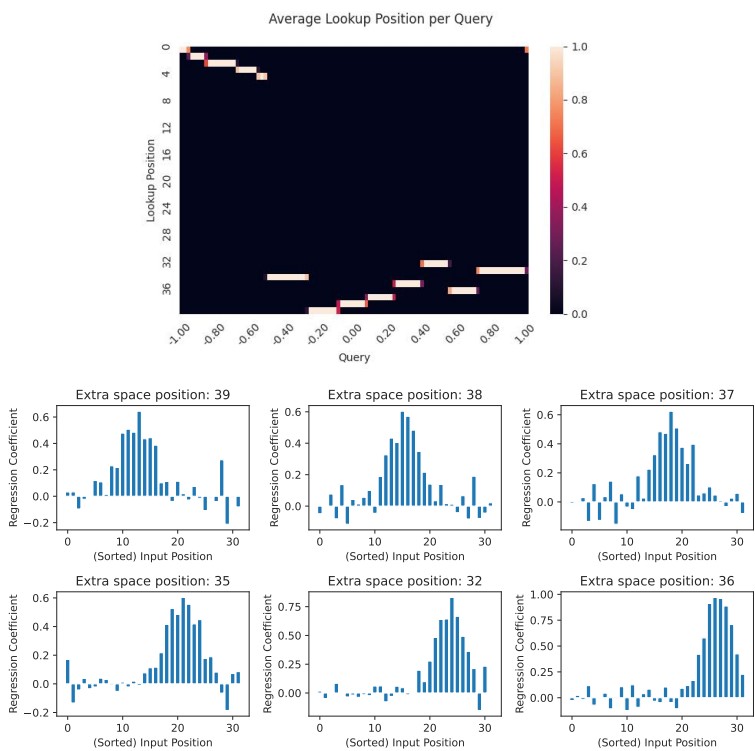

Figure 21: **(Top)** Decision boundary of the first query model. **(Bottom)** The regression coefficients of the values stored in extra positions as a linear function of the (sorted) inputs.

### B.12.2 Bucket Baseline

We create a simple bucket baseline that partitions $[-1, 1]$ into $T$ evenly sized buckets. In each bucket $b_i$ we store $argmin_{x_j \in D} ||x_j - l_i||$ where $l_i$ is the midpoint of the segment partitioned in $b_i$. This baseline maps a query to its corresponding bucket and predicts the input stored in that bucket as the nearest-neighbor. As $T \to \infty$ this becomes an optimal hashing-like solution.

### B.12.3 Understanding Extra Space Usage

By analyzing the lookup patterns of the first query model, we can better understand how the model uses extra space. In Figure 21 we plot the decision boundary of the first query model. The plot demonstrates that the model chunks the query space ($[-1, 1]$) into different buckets. To get a sense of what the model stores in the extra space, we fit a linear function on the sorted inputs and regress the values stored in each of the extra space tokens $b_i$ and plot the coefficients for several of the extra spaces in Figure 21. For a given subset of the query range, the value stored at its corresponding extra space is approximately a weighted sum of the values stored at the indices that correspond to the percentile of that query range subset. This is useful information as it tells the model for a given query

percentile how 'shifted' the values in the current dataset stored in the corresponding indices are from model's prior.

### B.12.4 30D Extra Space

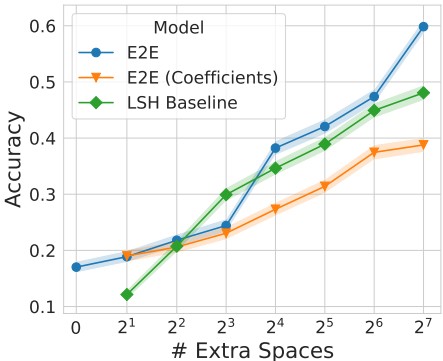

Figure 22: Our unconstrained model (E2E) and a more interpretable version (E2E (Coefficients)) both learn to effectively leverage an increasing amount of extra space in 30D, with the unconstrained model outperforming an LSH baseline.

In high-dimensions it is less clear what solutions there are to effectively leverage extra space, and in fact understanding optimal tradeoffs in this case is open theoretically [51].

We follow a similar setup to the 1D extra space experiments but use the data and query distributions from section 2.4. We experiment with two versions of extra space (unrestricted) and (coefficients). For the unrestricted version the data model can store whatever 30 dimensional vector it chooses in each of the extra spaces. For the coefficient model, instead of outputting a 30 dimensional vector, for each extra space, the model outputs a separate N dimensional vector of coefficients. We then take a linear combination of the (permuted) input dataset using these coefficients and store the resulting vector in the corresponding extra positions. While the unrestricted version is more expressive the coefficient version is more interpretable. We include both versions to demonstrate the versatility of our framework. If one is only interested in identifying a strong lower-bound of how well one can use a fixed budget of extra space they may use the unrestricted model. However, if they are more concerned with investigating specific classes of solutions or would like a greater degree of interpretability they can easily augment the model with additional inductive biases such as linear coefficients.

We plot the performance of both models along with an LSH baseline in Figure 22. While both models perform competitively with an LSH baseline and can effectively leverage an increasing amount of extra space, the unrestricted model outperforms the coefficient model at a certain point.

## C Frequency Estimation

### C.1 CountMinSketch

CountMinSketch [24] is a probabilistic data structure used for estimating the frequency of items in a data stream with sublinear space. It uses a two-dimensional array of counters and multiple independent hash functions to map each item to several buckets. When a new item $x$ arrives, the algorithm computes $d$ hash functions $h_1(x), h_2(x), \ldots, h_d(x)$, each of which maps the item to one of $w$ buckets in different rows of the array. The counters in the corresponding buckets are incremented by 1. To estimate the frequency of an item $x$, the minimum value across all counters $C[1, h_1(x)], C[2, h_2(x)], \ldots, C[d, h_d(x)]$ is returned. The sketch guarantees that the estimated frequency $\hat{f}(x)$ of an item $x$ is at least its true frequency $f(x)$, and at most $f(x) + \epsilon N$, where $N$ is the total number of items processed, $\epsilon = \frac{1}{w}$, and $w$ is the width of the sketch. The probability that the estimate exceeds this bound is at most $\delta = \frac{1}{d}$, where $d$ is the depth of the sketch (i.e., the number of hash functions). These guarantees hold even in the presence of hash collisions, providing strong worst-case accuracy with $\mathcal{O}(w \cdot d)$ space.

## C.2 Frequency Estimation Experiment Details

Here, we outline our approach to modeling the frequency estimation problem. Following NN search, we chose this problem to further explore the broader applicability of the end-to-end learning paradigm for the following reason: its streaming nature makes it fundamentally different from NN search, requiring adaptations to the framework used in NN search. In contrast, the other problems discussed in Section 3.2 require minimal adaptations (see App E). That said, similar to NN search, the two key constraints in this problem remain the size of the data structure and the number of lookups. As a result, the high-level setup remains the same: we still use a data-processing network and a query network, applying similar principles to optimize data structure efficiency. For instance, we control space complexity by tokenizing and restricting the data structure's size and enforce query complexity using similar sparsity techniques on lookup vectors. Next, we describe the data-processing and query networks. Note that while there are differences between the architectures used for NN search and frequency estimation, we explain how both problems fit into the broader framework in section E.

**Data processing Network** We model the data structure as a $k$ dimensional vector $\hat{D}$ and use an MLP as the data-processing network which is responsible for writing to $\hat{D}$. When a new element arrives in our stream, we allow the model to update M values in the data structure. Specifically, when an element arrives at time-step $t$, the data-processing network outputs $M$ $k$-dimensional update position vectors $u_1, ..., u_M$ and M corresponding scalar update values $v_1, ..., v_M$. We then apply the update, obtaining $\hat{D}_{t+1} = \hat{D}_t + \sum_{i=1}^{M} u_i * v_i$. Unlike in the NN setting where we did not constrain the construction complexity of the data structure, here we have limited each update to the data structure to a budget of $M$ lookups. We do so as in the streaming settings updates typically occur often, so it is less reasonable to consider them as a one-time construction overhead cost.

**Query processing Network** Query processing is handled in a similar fashion to NN search — we have $M$ query MLP models that output lookup positions. Finally, we also train a MLP predictor network $\psi(v_1, ..., v_M)$ that takes in the $M$ values retrieved from the lookups and outputs the final prediction.

**Training Details** We follow the same setup as the nearest neighbors training except for frequency estimation, the data-processing network is a 3-layer MLP with a hidden dimension of size 1024. We do a grid search over $\{0.0001, 0.00005, 0.00001\}$ to find the best learning rate for both models. Models are trained for 200k gradient steps with early stopping. All experiments are run on a single NVIDIA RTX8000 GPU.

## C.3 Improved Frequency Estimation with Augmented CountMinSketch

In our experiments on learning frequency estimation algorithms (Section 3.1), we found that on the Zipfian distribution our model was able to outperform the CountMinSketch algorithm by using a smaller update delta. We find that this can be particularly useful when the size of the data structure is small, and collisions are frequent. We hypothesize that the better performance of the learned solution is at least partially due to the smaller delta.

We use this insight to design a modified version of CountMinSketch that uses a custom update delta. In Figure 23, we show that this augmented CountMinSketch algorithm can outperform vanilla CountMinSketch on the large-scale CAIDA IP traffic dataset [25] by up to a factor of two. These results demonstrate that even at small scale, our model can provide useful insight into data structure design that can be transferred to realistic settings.

The CAIDA dataset [25] consists of traffic data collected in 2016 from a backbone link of a Tier-1 ISP between Chicago and Seattle. Each recording session spans approximately one hour, capturing around 30 million packets and 1 million unique flows per minute. We use the first minute for our experiment.

Recent work by Aamand et al. [37] independently arrived at a similar insight to the one our model discovered automatically. They propose a modified version of CountSketch that improves over standard CountSketch/CountMinSketch on Zipfian distributions, even without explicit predictions— relying only on knowledge of distributional skew. In comparison, our trained model learns to lower the update value uniformly for all elements, a strategy that mirrors the motivation behind their use

of conservative updates, where frequency estimates for some elements are set to zero. A theoretical comparison between the two approaches, as well as a deeper analysis of the learned update rule, is an interesting direction for future work. More broadly, the alignment between our automatically discovered strategy and their hand-crafted variant further supports the idea that small-scale learned models can yield transferable insights with potential for provable guarantees.

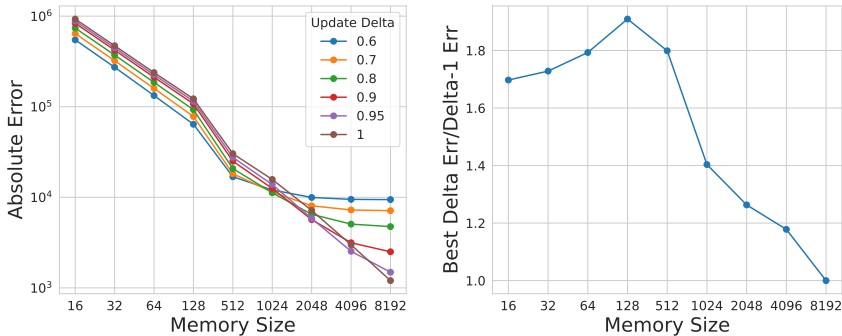

Figure 23: (**Left**) CountMinSketch performance on the CAIDA dataset with different update deltas vs. memory size. (**Right**) The relative performance of the best update delta vs the default delta ($\Delta = 1$) for different memory sizes. In some regimes, our augmented CountMinSketch can perform up to twice as well as vanilla CountMinSketch just by modifying the update delta.

## C.4  Learning Heavy Hitter Features

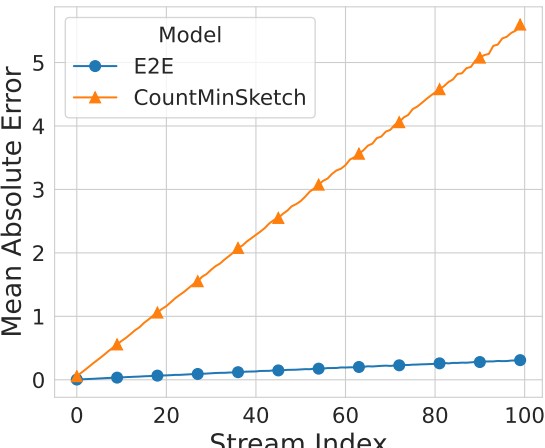

Figure 24: On the MNIST heavy-hitters frequency estimation experiment, our model significantly outperforms CountMinSketch. This is because our model can learn features predictive of heavy hitters, as opposed to the distribution-agnostic CountMinSketch.

In the previous experiment, the Zipfian distribution shape was fixed across training instances but the rank ordering of elements was random. In some settings, however, it may be possible to predict which elements are more likely to occur in the stream. While the exact elements may vary between streams, frequently occurring items could share features across streams. For instance, Hsu et al. [23] show that in frequency estimation for network flows, certain types of IP addresses receive much more traffic than others. We simulate such a setting by fixing the rank ordering of the Zipfian distribution. However, instead of using a universe of integer elements $\{1, ..., K\}$, we instead use their corresponding 3-digit MNIST images with $K = 100$ (constructed as in the MNIST NN experiment). Given a stream of integers, we map them to their corresponding MNIST labels and then for each label we sample a corresponding image from the training set. During evaluation, we use images

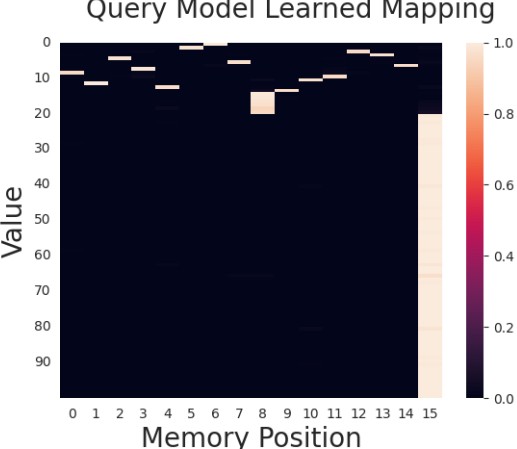

Figure 25: We show the decision boundary learned by the query/data-processing network in the MNIST heavy hitters experiment. As images with smaller numbers occur more frequently in the stream, the memory-constrained model learns to reserve separate memory positions for these items in order to prevent collisions among them.

samples from the test set. As the distribution is skewed and the ranking is fixed, images with smaller numbers are sampled much more frequently than those with larger numbers. As in the MNIST NN experiment, we also use a feature-learning CNN model to process the images before passing them to the data-processing and query-execution networks.

We compare our model to CountMinSketch with 1-query that is given the underlying labels instead of the images. Our model has a significantly lower error than the baseline (0.15 vs 2.81 averaged over a stream of size 100 (see Fig. 24)) as the latter is distribution-independent. By training from the data-distribution end-to-end, our framework is able to simultaneously learn features of heavy hitters (in this case, clustering images with the same label) and use this information to design an efficient frequency estimation data structure. We investigate the learned structure and find that the model has reserved separate memory positions for heavy hitters, thereby preventing collisions (Fig. 25).

## C.5 Relation of our experiments to Hsu et al. [23]

Here, we clarify the difference between our frequency estimation experiments and the setup explored by [23]. The frequency estimation experiment in the main paper (Figure 5 (Left)) is different from the work of [23] as we randomize the rank ordering of elements for each dataset so there are not consistent features of "heavy hitters" (frequently appearing elements). Rather, the models can only leverage the fact that the distribution is skewed (without knowing the frequency of any given element). However, in Appendix C.4 we include an experiment with MNIST data where the ranking is fixed but the model must learn features of heavy-hitters (some numbers appear more often than others). The model learns to do something similar to what [23] propose - using separate buckets to store counts for heavy hitters.

## C.6 Connection to neural sketching works [43, 44, 45]

The recent line of work on neural sketching algorithms [43, 44, 45] shares several similarities with our frequency-estimation experiments. Both can be regarded as memory-augmented neural networks that learn to read from and write to differentiable memory [38]. Although we did not run direct comparisons, their architectures and training setups—tailored specifically for frequency estimation—suggest performance comparable to, and potentially better than, ours, and likely offer a more scalable path for this task. By contrast, our goal is different: rather than building in inductive biases from known sketching algorithms to achieve scalability, we deliberately avoid such priors to ask a more fundamental question—can neural networks discover sketching algorithms from scratch?

### C.7 Frequency Estimation Failure Mode

In our frequency estimation experiments, we found a setting where learning struggled to make progress. Specifically, this occurs when the query and streaming distributions are Zipfian with a large skew. For fixed training instances, the rank order of elements is consistent across both stream and query distributions but randomized across different training instances.

We explain our hypothesis for why this occurs with a simplified setup. Imagine that the universe of elements that the query/streaming data is drawn from only consists of two elements: $\{A, B\}$. Assume the data structure size $k = 2$, thus the memory is large enough to store the counts for each element in separate buckets and therefore incur zero estimation error. Let $m_A = [x, y]$ and $m_B = [u, v]$ where $m_A$ denotes the initial lookup distribution for element $A$ and $m_B$ the distribution for $B$. In other words, when element $A$ arrives, the memory $\hat{D}$ is read by computing $m_A^T \hat{D}$ (and in a similar fashion for $B$). Now, consider the case when both $x > y$ and $u > v$ (this applies also when $x < y$ and $u < v$). If we sample a batch of datasets, in approximately half of them the frequency of $A$ will be much larger than the frequency of $B$ and in the other half the inverse will be true. Again, this is because while the degree of skew is fixed across datasets, the rank ordering of elements is randomized. Consequently, this means that for data sets where $A$ is more frequent than $B$, the optimizer will try to increase $x$ and lower $y$ and when the inverse is true, the optimizer will try to increase $u$ and lower $v$. However, this means that for both elements, the lookup positions will point to the first position in memory over the second (the opposite is true when $x < y$ and $u < v$). This will cause collisions and increase the estimation error. This behavior can be avoided by setting $v > u$, or more generally, making the lookup distributions more orthogonal.

While this is a relatively simplified setup (i.e. we're not using any neural networks), we believe it captures part of the more general phenomenon. It demonstrates that there are settings where careful initialization can be required for learning to make progress. More generally, this suggests that end-to-end learning cannot always be expected to work without additional problem specific inductive biases. In our view, it also makes it more surprising that for the majority of settings we explored, we did not encounter such issues!

## D Efficiency & Scaling

### D.1 Improving Efficiency of Data-Processing Network for NN Search

For our NN experiments, we chose to use a transformer with $O(n^2)$ complexity in order to keep the model relatively general. This may not always be necessary. For instance, high-dimensional hashing-based methods could be learned with a less complex model. However, data structures like k-d trees have a higher construction complexity $O(dn \log(n))$. Our framework does not require using any specific model so a less (or more) complex model can trivially be substituted in for the transformer. One way to enable scaling to larger instances would be to lower the complexity of the data-processing model (e.g. using a linear transformer).

To demonstrate that in some settings quadratic attention can be substituted with a cheaper alternative, we run additional experiments in 2D (uniform distribution) and in high dimensions (FashionMNIST) with the linear attention Performer model [52].[7] We use the same model hyper-parameters as the quadratic attention model and plot the results in Figure 26. The comparable performance of the linear attention model suggests that it could be a computationally-cheaper alternative that can enable scaling up models to larger settings.

### D.2 Scaling Up

Most of our nearest neighbor search experiments are done with input dataset sizes around $N = 100$, however, we are also able to scale up to $N = 500$ (Figure 27 (Left/Center)), though with less than $\log(N)$ queries. There are two primary limitations that led to relatively low $N$: the data-processing cost (since all $N$ data points are processed) and the MLP query models, which output vectors of size $N$—causing heavy matrix multiplication and memory costs. While we demonstrate that useful data

---

[7]We use the Pytorch implementation from https://github.com/lucidrains/performer-pytorch/tree/main.

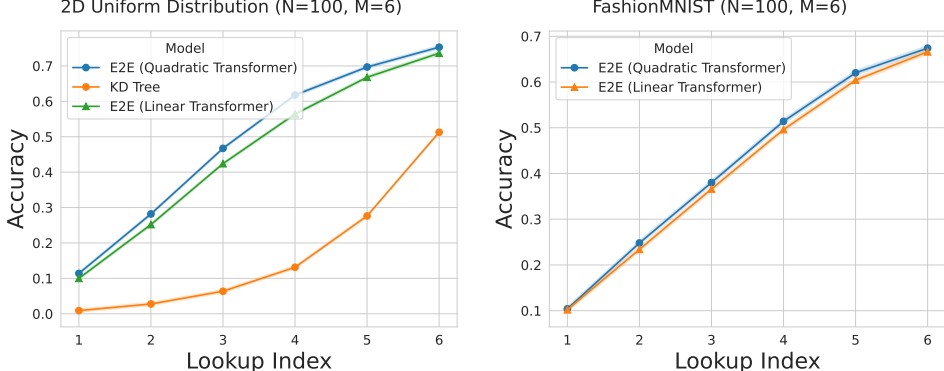

Figure 26: Performance of the quadratic attention transformer vs. linear attention performer transformer [52] in both 2D (**left**) and high dimensions (100) (**right**).

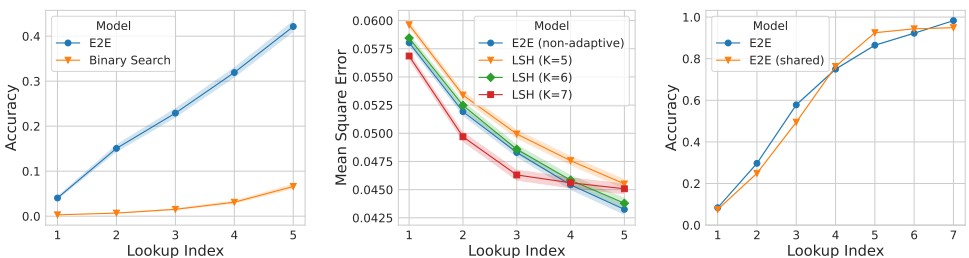

Figure 27: We scale both the 1D (**Left**) and 30D (**Center**) experiments to datasets of size $N = 500$. (**Right**) We compare our E2E model with a version where the query-execution network is only composed of one query-model (E2E (shared)) that is used in a loop for $M = 7$ queries during training on the 1D Uniform distribution, thereby conserving parameters by reusing weights. This could be a promising direction for problem settings where there is a recursive structure to the query algorithm.

structures can still be learned at this scale, it is possible that other classes of structures only emerge for larger datasets.

One avenue to scale end-to-end learning to larger datasets is by increasing the parameter count of the data-processing and query-execution networks. Moreover, as transformers become increasingly efficient at handling larger context sizes in language modeling settings, some of these modeling advancements may also be used for scaling models in the context of data structure discovery. While helpful, these changes would require resources beyond those available with academic compute.

Complementary to our work, it could also be valuable to explore better inductive biases for the query and data-processing networks, and other methods to ensure sparse lookups, enabling smaller models to scale to larger datasets. For instance, using shared weights among query models can be helpful in scaling up the number of queries. As a first step in this direction we show that a single query model can be used in-a-loop for NN search in 1D (Figure 27 (Right)). We leave further investigation for future work.

Another inductive bias that could improve scaling would be some form of hierarchical indexing. For example, the data-processing model could assign data points to partitions in a hierarchy, while the query network outputs a smaller set of coordinates to index this hierarchy.

These are just initial ideas and there is a large space for inductive biases worth exploring in future works. We deliberately avoided adding such inductive biases as we wanted to leave the setup relatively general and see if anything useful could be learned even at smaller scales. We also emphasize that many of the insights that can be derived from our models' learned solutions would scale to larger $N$. For instance, k-D trees in 2d and locality-sensitive hashing in higher dimensions. We limit ourselves

to datasets of these sizes due to computational constraints, and because our primary goal was to understand whether end-to-end data structure design is feasible at any reasonable scale.

Finally, we point to other works focused on algorithm learning and discovery that started at small scale with more synthetic setups [2, 53, 54] and inspired follow-up work that then focused on scaling up the initial set of ideas. We hope our work can serve a similar purpose - demonstrating the value in learning data structures E2E - and inspire future work to develop methods for better scaling.

### D.3 Benchmarking Computational Overhead

In this section we evaluate the throughput of the nearest neighbor models across various settings. We emphasize that in this paper we did not optimize for practical efficiency as we were primarily interested in the fundamental tradeoffs between space and query complexity, though these are certainly related to runtime in practice. Nevertheless, we include our findings for completeness. We use the same architectures from our experiments (see App B.1 for details). Benchmarking is conducted with a batch size of 512 and the results are averaged over 10 trials on a single GPU.

#### Query Processing Throughput

The table below shows the number of queries processed per second when using $\log(n)$ lookups per query. As expected, querying smaller datasets has higher throughput as the number of lookups, and thus neural net forward inferences, is smaller.

| Dim | $N = 64$ | $N = 128$ | $N = 256$ | $N = 1024$ | $N = 2048$ |
|---|---|---|---|---|---|
| 32 | 212,943 | 175,559 | 146,495 | 113,960 | 103,476 |
| 64 | 211,518 | 173,807 | 149,603 | 113,249 | 105,833 |
| 128 | 208,741 | 172,275 | 151,533 | 116,607 | 104,858 |

Below, we fix $N = 256$ and measure how increasing the number of lookups ($M$) affects throughput. There is a clear inverse correlation between $M$ and throughput indicating that query complexity has implications for practical efficiency.

| Dim | $M=1$ | $M=2$ | $M=3$ | $M=4$ | $M=5$ | $M=6$ | $M=7$ | $M=8$ |
|---|---|---|---|---|---|---|---|---|
| 32 | 1,072,476 | 517,067 | 345,946 | 261,705 | 212,202 | 176,503 | 145,927 | 134,165 |
| 64 | 1,090,290 | 513,129 | 355,654 | 265,753 | 211,991 | 174,031 | 150,917 | 132,594 |
| 128 | 1,054,366 | 519,691 | 348,347 | 260,985 | 212,237 | 173,406 | 145,191 | 133,278 |

#### Data-Preprocessing Throughput

The table below shows the number of raw datasets processed into a data structure per second using quadratic and linear attention. Quadratic attention runs faster here due to flash-attention [55], which optimizes memory access. Notably, in smaller $N$ regimes, linear attention can be less efficient than quadratic attention with hardware optimizations, despite its lower FLOP count.

| Arch | Dim | $N = 64$ | $N = 128$ | $N = 256$ | $N = 1024$ | $N = 2048$ |
|---|---|---|---|---|---|---|
| Linear | 32 | 17,701 | 8,772 | 4,542 | 1,170 | 585 |
| Linear | 64 | 17,241 | 8,844 | 4,563 | 1,177 | 590 |
| Linear | 128 | 17,121 | 8,614 | 4,473 | 1,152 | 577 |
| Quadratic | 32 | 95,229 | 49,697 | 13,418 | 5,867 | 571 |
| Quadratic | 64 | 94,815 | 46,924 | 13,285 | 5,248 | 548 |
| Quadratic | 128 | 93,294 | 48,506 | 13,095 | 5,174 | 543 |

We also try fixing the input dimension to 128 and adjust the number of layers in the quadratic transformer.

| # **Layers** | $N = 64$ | $N = 128$ | $N = 256$ | $N = 1024$ | $N = 2048$ |
|---|---|---|---|---|---|
| 4 | 174,433 | 174,381 | 173,501 | 31,512 | 8,449 |
| 8 | 94,493 | 95,037 | 18,849 | 6,780 | 610 |
| 12 | 65,331 | 24,301 | 8,073 | 1,103 | 220 |

# E    Broader Applicability of Our Framework

## E.1    Generality of Our Framework

Our framework is designed for problems that require efficiently answering queries over some collection of data. We believe it can be applied to any problem that shares this structure—namely, one where there is a data-processing step, a query algorithm, and constraints on space and query-time complexity. In Section 3.2, we outline several such candidate problems that could benefit from this approach. Many classical data structure problems share these common elements, and our framework provides a unified way of modeling them.

A key feature of this formulation is the enforcement of query-time complexity constraints, which requires learning sparse querying algorithms. We address this by training with noise to promote sparsity (see App. B.3), a core mechanism that makes the framework practical in constrained settings.

In its most general form, the framework consists of three components: a data-processing network that maps $N$ data points into $K$ structured chunks; a query model that performs $M$ lookups over these chunks based on a given query; and a predictor network that produces the final response.

In nearest neighbor search, we have $N = K$, and the inputs must be retained during data structure construction. This motivates learning permutations as part of the data model. As discussed in App. E.2, this approach does not limit generality and can apply to any setting where inputs must be preserved. Additionally, because the final response is itself a data point, no separate predictor network is needed.

In contrast, frequency estimation introduces a structurally different challenge: $N > K$ (stream length vs. memory size), and the data-processing network uses an MLP instead of a Transformer to better suit the streaming setting. Here, we also include a predictor network to estimate frequencies, while reusing the same query mechanism from the NN setup. We chose frequency estimation as our second problem precisely because it is structurally different due to the streaming nature, yet E2E learning still works under the proposed framework.

**Adapting Framework for Other Problems**

As nearest neighbour search is arguably the most pervasive data structure problem, there remain many other avenues to explore end-to-end learning for nearest neighbour search beyond the settings we discuss in our paper, including $k$-NN search, different distance metrics such as $L_1$ distance, and structures that can support insertions and deletions. Moreover, the frequency estimation setting could be easily adapted to cover membership data structures such as bloom filters.

For new data structure problems, such as those we mention in Section 3.2, one could certainly use the same architectures we have used, i.e. for the data-processing model, using a transformer for the batch-setting or MLP for the streaming setting and using MLPs for the query-model. These architectures are relatively general and therefore require minimal adaptation. To be more specific, we discuss how each of the problems we propose in Section 3.2 can be tackled with our framework. We focus on the batch setting as this is the more common one though similar insights apply to the streaming setting.

**Graph data structures**    A permutation-equivariant architecture such as a transformer (with no positional encodings) or a graph-neural network should be used to represent the graph in the data-processing network. Note that it has already become commonplace to use transformers to represent graphs [56, 57]. The output size of the model would be constrained to control the space complexity of the learned data structure. This is accomplished by either ignoring certain output tokens when less space then the input size is desired or adding additional tokens when more space is required, as we do in our extra space experiments. The same MLP query architecture we proposed should suffice.

**Sparse matrices**   As matrices are simply grids of numbers, transformer would be appropriate as well with each token representing either a row/column of the matrix or a single element. The output size of the model would be constrained to control the space-complexity of the learned data structure. The same MLP query architecture we proposed should suffice.

**Learning statistical models**   In this case, the data-processing network which operates over datasets should likely be permutation-invariant as there is no canonical ordering to an IID dataset so a transformer would suffice. Again, the output size of the model would be constrained to control the size of the learned model (e.g. decision tree). The MLP query architecture would be an appropriate choice for the query model.

While specialized architectures do exist for different domains (e.g. graph neural networks), transformers can certainly be used as a starting point and are currently applied to many different types of inputs [58, 59, 56]. There is also work that is trying to build general architectures for structured inputs/outputs [60, 61]. Advances in this direction could lead to more general architectures for data structures as well

In summary, our framework is general in so far as it provides a concrete way to set up the interaction between the data structure and the queries while respecting space and query complexity constraints. It does not require prior knowledge of specific data structures or algorithms. That said, this should be viewed as a starting point—not an all-encompassing recipe—and using problem-specific information can certainly improve performance.

### E.2   Nearest Neighbor Permutation

A key NN constraint is that the prediction must be one of the original data points, which every NN method (even approximate ones) satisfies. Initially, we trained a transformer to output a re-ordered dataset directly, but training was slower because it had to recreate the original data as well as reorder it. Instead, we trained the model to learn a permutation directly, avoiding the need to learn to reproduce the data while still preserving expressivity, since any structure (e.g. clustering or hierarchical ordering) can be represented as a permutation. Our experiments, such as the FashionMNIST plot in Figure 18, confirm that this method naturally discovers clustering. Beyond NN, it can benefit any task requiring retention of the inputs.

