# OpenReview forum: "Discovering Data Structures: Nearest Neighbor Search and Beyond"
_NeurIPS.cc/2025/Conference — NeurIPS 2025 poster_

### Official Review · Reviewer_ffsc · 2025-06-28

**Clarity:** 4
**Significance:** 4
**Originality:** 4
**Rating:** 5
**Confidence:** 3

**Summary:**

The paper proposes an end-to-end framework for data structure training for NN search tasks and beyond. The method comprises a differential sorting network to form the data structure and a query network to conduct the query. This framework works great in different NN search tasks with different data dimensions and distributions. The paper also explores its application on tasks beyond the NN search like frequency estimation.

**Questions:**

See weakness above.

**Ethical Concerns:**

["NO or VERY MINOR ethics concerns only"]

**Final Justification:**

This is a solid and interesting work, and the authors' responses have addressed most of my concerns. Thus, I maintain my accept score.

**Limitations:**

yes.

**Paper Formatting Concerns:**

No Concerns

**Quality:**

4

**Strengths And Weaknesses:**

Strengths:
1. The paper writing is fluent and easy to read, the motivation is easy to follow and the figures and tables are also clear.
2. The method design is simple and effective, forming the data structure on the fly when training on query tasks.
3. The experiment findings are very interesting, which forms different structures in different data dimensions and the structures align with some data structure algorithms. These findings also show the effectiveness of the method.
4. The method can be used in tasks beyond NN search, like frequency estimation, and also show promising results.
5. The experiments are very sufficient to prove the claims of the paper.

Weaknesses:
1. Could you describe how to adapt your model in frequency estimation in detail? I still don't understand the pipeline of this application.
2. In the current age, the dimensions of 30 and 100 are still a little bit small. Have you tried your method on more dimensions?
3. If the data processing model can try to struct the data not only in a 1D sorting, but in a 2D map or higher dimension, the learnt data structure may have more interesting properties and have more further applications.

---

> ### Author Rebuttal · Authors · 2025-07-30
>
> We thank the reviewer for your feedback and appreciate the time and effort you’ve dedicated to evaluating our work!
>
> >Could you describe how to adapt your model in frequency estimation in detail? I still don't understand the pipeline of this application.
>
> Unfortunately, due to space constraints we were unable to provide more detail in the main body. However, in App C.2 we provide a more detailed description of how we adapt the model for frequency estimation. In Appendix E, we discuss the broader applicability of the framework and how it can be adapted to several other problems. Please let us know if there are any specific details that are still unclear and we’d be happy to respond!
>
> >In the current age, the dimensions of 30 and 100 are still a little bit small. Have you tried your method on more dimensions?
>
> Yes, we include experiments with a 3-digit version of MNIST which has dimensions of 28*84 in Section 2.4. We find that the model learns a low-dimensional representation of the data, processes the data into a data structure and queries the data structure all in-tandem end-to-end.
>
> >If the data processing model can try to struct the data not only in a 1D sorting, but in a 2D map or higher dimension, the learnt data structure may have more interesting properties and have more further applications.
>
> We definitely agree! As both models are being trained end-to-end the data-structure model effectively learns a useful space partitioning in high-dimensions that can be efficiently queries. For instance, in Figure 18 we show the model’s learned clustering of FashionMNIST data (100 Dimensions).
>
> We hope this response answers your questions and addresses your concerns. Please let us know if you have any other questions!

---

> ### Comment · Reviewer_ffsc · 2025-08-01
>
> 1. For the frequency estimation:
>
> In fact, I have already carefully read the appendix in App C.2 though, I still don't fully understand, so I proposed this question. I think you could explain about what's the inputs and outputs mean in both your Data processing Network and Query processing Network in this case and how to construct the frequency based on the output.
>
> (Note: I understand that in NN search, inputs of Data processing Network are the data in database and outputs are their features, and the input of Query processing is query and output is the position. The position is the key to find the NN search position to solve this problem)
>
> 2. For the more applications:
>
> I ask this question, to be honest, means that I hope to see some applications using these interesting learned features. For example, do the features have representative ability? You can test this by classification using linear probing to test the representative ability of this feature. Or do the features have generalizable ability? After training on 3-digit MNIST, what's the performance of it applying to 1 / 2-digit hand-writing numbers?
>
> Overall, I consider this a solid and well-executed work. The questions raised above were primarily intended to help further improve the paper's completeness.

---

> > ### Author Response · Authors · 2025-08-01
> >
> > Thanks for clarifying your questions and we appreciate your interest in our work and kind comments! Please find our responses below.
> >
> > >In fact, I have already carefully read the appendix in App C.2 though, I still don't fully understand, so I proposed this question. I think you could explain about what's the inputs and outputs mean in both your Data processing Network and Query processing Network in this case and how to construct the frequency based on the output.
> >
> > To better explain the setup, we’ll use a concrete example. Imagine a stream S of 5 numbers, each arriving one at a time:
> >
> > S(1) = 2
> >
> > S(2) = 3
> >
> > S(3) = 7
> >
> > S(4) = 1
> >
> > S(5) = 2
> >
> >
> > For frequency estimation, the input to the data-processing network is an element from the stream whose count we want to update, for instance, at timestep 1 the element is 2, at timestep 4 the element is 1, etc. The data-processing network needs to update the memory to update the count of the element. For example, at the start of the stream all counts should be 0 and after timestep 1, the count of element 2 should be 1.
> >
> > The input to the query network is the element whose count we want to query and the query network should output the count of that element. For instance, before the stream, the query network should output 0 for all elements, after the stream, the query network should output 1 for element 1, 2 for element 2, 1 for element 3, 1 for element 7, and 0 for any other element.
> >
> > To be more specific, we can describe exactly how the data-processing model writes from the memory and the query network reads from the memory at a given timestep in a simple single-query setting.
> >
> > Let $\mathbf{D}_0$ be a $k$-dimensional zero-initialized vector representing the initial memory at the start of the stream. At timestep $t$ the data-processing network handles updates as follows:
> >
> > In the single query setting, the data-processing model, which is an MLP takes in an element (scalar number in this setting) and outputs a $k$-dimensional vector $\mathbf{u}$ as well as a scalar value $v$. The vector m represents a one-hot encoded address, just like in the nearest neighbor setting. We then update the memory with the operation: $\mathbf{D}_{t+1} = \mathbf{D}_t + \mathbf{u}*v$.
> >
> > To read from the memory at time-step $t$, the query model (also an MLP), takes an element e and outputs its address $\mathbf{u}$. We then read an output $o$ from the memory by computing $o=<\mathbf{D}_t, \mathbf{u}>$. Finally, we have a small MLP predictor model that takes in $o$ and is trained to output the correct count of element $e$ by regression.
> >
> > The multi-query setting generalizes this by allowing the data-processing model to update $M$ positions at a time and the query-processing network can read from $M$ positions, deriving M outputs $o_1,...,o_M$. The predictor model then takes these $M$ outputs and predicts the final count.
> >
> > We designed the setup to be relatively general in order to add very little inductive bias around how the memory should be structured as we wanted to see if the model could learn useful algorithms by itself. In Appendix E we discuss how both nearest neighbor search and frequency estimation fit into the broader framework of data structure discovery.
> >
> > >I ask this question, to be honest, means that I hope to see some applications using these interesting learned features. For example, do the features have representative ability? You can test this by classification using linear probing to test the representative ability of this feature. Or do the features have generalizable ability? After training on 3-digit MNIST, what's the performance of it applying to 1 / 2-digit hand-writing numbers?
> >
> > We found that the learned features correspond well with the 3-digit labels up to a linear transformation, see Figure 4 (Center). Also, the trained feature model generalizes in-distribution in the sense that at test-time we give it test images drawn from the same distribution but not seen during train time and the model still performs well. While this is standard behavior in image classification, we still found it surprising the model was able to learn generalizable features while still handling the data structuring and querying objectives. We didn’t test the out-of-distribution generalization of the learned feature model, e.g. by giving it 2-digit numbers instead of 3-digit numbers. It’s possible that the model can still classify these 2-digit numbers given that they’re already a subset of the 3-digit numbers and we’re using a CNN so it can scale to smaller images without retraining.
> >
> > In the next draft, we'll provide these extra details in order to improve the clarity of the paper. We hope this answers your questions, please let us know if you have any other ones!

---

> > > ### Comment · Reviewer_ffsc · 2025-08-02
> > >
> > > Thank you for your response. I have no further questions then. This is an interesting and solid work. I’m inclined to maintain my original accept rating.

---

### Official Review · Reviewer_LJYb · 2025-06-29

**Clarity:** 3
**Significance:** 2
**Originality:** 2
**Rating:** 3
**Confidence:** 5

**Summary:**

This paper investigates the feasibility of learning data structures end-to-end using neural networks, with a primary focus on the nearest neighbor search problem in both low and high dimensions. The authors propose a novel framework consisting of two jointly trained components: a data-processing network that constructs a data structure from raw input data, and a query network that learns how to query it efficiently.
The study demonstrates that, neural networks can rediscover classical algorithms such as sorting followed by binary search, and in higher dimensions, structures resembling k-d trees or locality-sensitive hashing. The framework is extended to a frequency estimation task in data streams, showcasing its broader applicability. The paper highlights both the potential and challenges of this learning paradigm and provides empirical evidence that neural models can autonomously learn interpretable and effective data structures from scratch.

**Questions:**

1.  Please address W1 by discussing relevant prior work on end-to-end neural data structures learning, and explaining how the proposed approach differs from or improves upon these works. It would also be helpful to include experimental comparisons to highlight the novelty or advantages of the proposed method.

2.  Please support or clarify W2 with additional experiments, preferably by providing efficiency-related metrics such as runtime, throughput, or similar comparisons.

3.  For W3, it would be helpful if the authors could include empirical measurements of preprocessing and training time across different model sizes, as well as how these metrics scale with the size of the dataset.

4.  For W4, although the authors mention that worst-case guarantees cannot be provided, it would still be valuable if they could offer any empirical observations or partial theoretical guarantees.

**Ethical Concerns:**

["NO or VERY MINOR ethics concerns only"]

**Final Justification:**

My primary concern with this paper is that it overlooks two important lines of related work (the CLRS benchmark and Neural Sketches) , which were not discussed or experimentally compared at all in the original version. This omission poses a significant challenge to both the completeness and the novelty of the work.

While I accept that the CLRS benchmark need not be included in experiments due to substantial technical differences, it should still be discussed as representative work in the field. The omission of Neural Sketches is far more critical: despite acknowledging its close technical and conceptual similarity to this paper, the authors argue it should not be compared, citing a focus on nearest-neighbor search. However, this task is inherently supported by memory-augmented neural networks, a capability central to the Neural Sketches series of works. This omission undermines the paper’s completeness and scholarly positioning, and cannot be addressed within the scope of a camera-ready version.

**Limitations:**

Yes

**Quality:**

2

**Strengths And Weaknesses:**

Strengths：

1. The paper introduces an interested idea—treating data structure design as an end-to-end learning problem.

2. The proposed framework is effectively applied to classic tasks such as nearest neighbor search and frequency estimation. It can outperform classical data structures under uniform distribution and demonstrate comparable performance under hard distribution.

3.The authors provide thorough ablations that isolate the contributions of different components, accompanied by well-reasoned explanations.

4.The paper is well written and clearly organized.

Weaknesses：

1. W1(Related Works Missing). The paper seems to miss some important prior work on end-to-end learning of data structures. As far as I know, [1] introduced the CLRS Algorithmic Reasoning Benchmark, which also trains graph neural networks in an end-to-end manner to perform dozens of classical algorithms such as sorting and searching. That work, along with follow-up studies like [2, 3], should have been mentioned. In addition, for the task of frequency estimation, there are also recent neural data structure approaches based on end-to-end learning, such as [4, 5, 6]. These cutting-edge and highly relevant works are not discussed or compared against in the paper, which feels like a significant oversight.

2. W2(Efficiency Overheads). The approach often relies on large Transformer-based architectures, leading to significant pre-processing and inference costs. The quadratic complexity of standard attention limits applicability without further optimization.

3. W3(Scalability Limitations). Most experiments are conducted on small datasets, raising concerns about the framework’s feasibility for large-scale or real-world applications. Scaling to practical-sized problems remains an open challenge.

4. W4(Guarantee Absence). Despite the strong empirical results, the absence of worst-case guarantees is a notable limitation—especially important when comparing to well-understood classical algorithms.

[1] The CLRS Algorithmic Reasoning Benchmark. ICML

[2] Neural Algorithmic Reasoning with Causal Regularisation. ICML

[3] Neural Algorithmic Reasoning Without Intermediate Supervision. NeurIPS

[4] Meta-sketch: A Neural Data Structure for Estimating Item Frequencies of Data Streams. AAAI

[5] Learning to Sketch: A Neural Approach to Item Frequency Estimation in Streaming Data. TPAMI

[6] Lego Sketch: A Scalable Memory-augmented Neural Network for Sketching Data Streams. ICML

If these issues can be resolved, I will consider further improving my score.

---

> ### Author Rebuttal · Authors · 2025-07-30
>
> Thank you very much for your detailed review and for providing useful feedback!
> ## W1: Relation to Prior Work
> Thanks for sharing these references! We comment on them below and will definitely include a more thorough discussion on these in our next draft.
>
> **CLRS**
>
> While works related to the CLRS benchmark [1, 2, 3] share a similar spirit to ours of studying neural networks (NNs) and classical algorithms there are several key differences:
> - These works train NNs to simulate specific algorithms (e.g. bellman-ford). We aren’t trying to learn any specific algorithm but rather want the learned algorithms to satisfy certain conditions (i.e. space and query complexity constraints).
> - Our setup uses significantly less supervision than these works which usually provide the model with intermediate outputs of the algorithm they’re trying to learn as supervision. While [3] does not use this supervision they do add additional inductive biases such as a contrastive learning objective to make the problem more tractable.
> - Our setup is more challenging as the model needs to learn two algorithms in tandem (data processing and query algorithms) from scratch.
> - These works tend to use specialized graph NNs whereas we instead introduce minimal problem-specific biases, aiming to test whether general-purpose NNs can learn data structures.
>
> **Neural Sketching**
>
> We were not aware of this recent work on neural sketching algorithms and thank the reviewer for pointing it out! Their approach shares several similarities with our frequency estimation experiments—both can be viewed as memory-augmented neural networks trained to read from and write to differentiable memory [8]. While we did not run direct comparisons, their architectures and training setups suggest performance comparable to, and possibly better than, ours, as they were tailored specifically for frequency estimation. In contrast, our experiments aimed to show the generality of our framework, with our main focus on nearest-neighbor search.
>
> We also highlight a difference in motivation. Their work targets scalable, practically efficient neural sketches, whereas ours emphasizes theoretical notions of efficiency, such as query complexity. For instance, while their work encourages sparse read/writes we carefully enforce one-hot query sparsity in order to control the number of lookups. This design choice lets us draw direct comparisons to classical sketches (e.g. Count‑Min Sketch) in terms of query complexity, rather than solely throughput or runtime.
>
> Finally, while their work focuses on frequency estimation, our work is more concerned with identifying the broader principles behind data structure discovery than trying to learn practical data structures for a single problem. For instance, it’s not clear how to translate their framework to also handle NN search or some of the other problems we discuss. We will include these references in the next draft, discuss their connections to our work, and either provide a comparison or acknowledge that their method likely performs similarly or better for this task.
> ## W2/W3: Efficiency & Scale
>
> We definitely understand the concerns around the efficiency and scalability of our methods and acknowledge that it’s necessary for future work to address this in order to build a system suitable for production workloads. Our current methods are not designed to scale to large datasets and we believe that more problem-specific inductive biases are crucial for this (see App D for a discussion).
>
> We’d like to emphasize, however, that **our intention with this work was not to propose a method built for real-world deployment. Instead, we are interested in understanding to what extent neural networks can explore the space of data structures—even at small scale—from a classical algorithms perspective.**
>
> Initially, it was unclear (at least to us) whether end-to-end learning of data structures for tasks like nearest neighbors was even feasible, or how to even frame the problem. It raises many fundamental questions: How should data structures and queries be represented using neural networks, especially under combinatorial constraints like limited query budgets? Will training succeed at all, given that the data-processing and query networks are trained from scratch and rely entirely on each other? Even then, can such a system explore the space of classical and novel algorithms?
>
> To our surprise, once we carefully set up training—with space and query constraints and a structured lookup interface—the model was able to discover meaningful algorithms! It recovered classic data structures like sorting, KD-trees, and LSH-like behavior, often adapting them to the underlying distribution.
>
> To better understand what the model is learning, we place a strong emphasis on reverse-engineering its behavior and comparing it to classical algorithms. This perspective also motivates our focus on theoretical notions of efficiency such as query and space complexity—quantities we can directly enforce and optimize during training. In contrast, metrics like wall-clock time are often closely tied to hardware and implementation details and were not the central focus of our investigation.
>
> To summarize, we view this as our core contribution: framing data structure discovery as an end-to-end learning problem and showing that it can explore a rich algorithmic space.  Finally, to our knowledge, this is one of the first attempts to approach data structure discovery in a general way. While our experiments focus on NN search, the design principles extend to other algorithmic settings. Of course, scaling to real-world workloads is an exciting next step, but our results show that even small-scale models can yield useful insights—potentially informing the design of scalable, hand-engineered systems
>
> We hope this clarifies our intentions with this work. Nevertheless, we’ve measured the throughput of our models for the nearest neighbor experiments in several regimes.
>
> ### Performance Metrics
> We emphasize that we did not optimize for practical efficiency as we were primarily interested in the fundamental tradeoffs between space and query complexity, though these are certainly related to runtime in practice. We use the same architectures from our paper (see App B.1 for details). Benchmarking is conducted with a batch size of 512 and the results are averaged over 10 trials on a single GPU.
>
> **Querying Throughput**
>
> Number of queries processed per second when using log(n) lookups per query. As expected, querying smaller datasets has higher throughput as the # of lookups, and thus neural net inferences, is smaller.
>
> | dim  | n=64   | n=128  | n=256  | n=1024 | n=2048 |
> |------|--------|--------|--------|--------|--------|
> | 32   | 212943 | 175559 | 146495 | 113960 | 103476 |
> | 64   | 211518 | 173807 | 149603 | 113249 | 105833 |
> | 128  | 208741 | 172275 | 151533 | 116607 | 104858 |
>
> **Data-preprocessing Throughput**
>
> The table below shows # datasets processed into a data structure per second using quadratic and linear attention (which we explored as a faster alternative in App. D.1). Quadratic attention runs faster here due to flash-attention [7], which optimizes memory access. Notably, in smaller N regimes, linear attention can be less efficient than quadratic attention with hardware optimizations, despite its lower FLOP count [7].
>
> | Arch         | Dim | n=64  | n=128 | n=256 | n=1024 | n=2048 |
> |--------------|-----|-------|-------|-------|--------|--------|
> | **Linear**   | 32  | 17701 | 8772  | 4542  | 1170   | 585    |
> | Linear       | 64  | 17241 | 8844  | 4563  | 1177   | 590    |
> | Linear       | 128 | 17121 | 8614  | 4473  | 1152   | 577    |
> | **Quadratic**| 32  | 95229 | 49697 | 13418 | 5867   | 571    |
> | Quadratic    | 64  | 94815 | 46924 | 13285 | 5248   | 548    |
> | Quadratic    | 128 | 93294 | 48506 | 13095 | 5174   | 543    |
>
> We also try fixing the data dimension to 128 and adjust the # of layers in the quadratic transformer.
>
> | # Layer | n=64   | n=128  | n=256  | n=1024 | n=2048 |
> |---------|--------|--------|--------|--------|--------|
> | 4       | 174433  | 174381 | 173501 | 31512  | 8449   |
> | 8       | 94493  | 95037  | 18849  | 6780   | 610    |
> | 12      | 65331  | 24301  | 8073   | 1103   | 220    |
>
> **Train Time**
> Below is the train time for each of our experiments with N=100 sized datasets
> | 1D        | 2D        | 30D        | MNIST      | SIFT       | FashionMNIST |
> |-----------|-----------|------------|------------|------------|--------------|
> | ~3 Hours  | ~3 Hours  | ~10 Hours  | ~10 Hours  | ~2 Hours   | ~2 Hours     |
>
> ## W4: Guarantees
> Thanks for raising this point! While our learned data structures lack formal worst-case guarantees, their empirical performance is encouraging and can provide some insight:
> - We find that in many regimes, the models learn algorithms resembling classical ones—e.g., binary search, interpolation search, KD trees, and LSH—in both structure and querying, indicating they’re approximating the same performance guarantees those algorithms enjoy.
> - Beyond 1D search, few methods offer worst-case guarantees better than O(n); for example, both LSH and KD trees have O(n) worst-case complexity. Our models can trivially match this by exhaustively searching, but more meaningfully, we show that their average-case performance is often comparable or better.
> - One could use our setup to derive insights that could then be translated into a learning-augmented algorithm that does have guarantees (see App C.3. for an example with frequency estimation), or combine it with an existing approach, for instance use a learned model for a quick search and fall back to a method with guarantees if nothing is found, thereby getting the best of both worlds.
>
> We hope this response addresses your concerns. Please let us know if you have other questions!
>
> [7]  https://arxiv.org/abs/2205.14135
>
> [8] https://www.nature.com/articles/nature20101

---

> > ### Comment · Reviewer_LJYb · 2025-08-04
> >
> > Thank you for your reply. I appreciate your discussion of the two missing lines of work. We both agree that these works are directly tied to the core contribution of the paper and should be incorporated into the discussion and comparison to make the paper more complete and better aligned with the title “End-to-End Learning for Data Structures.”
> >
> > Therefore, I hope that the discussions and comparative experiments related to the missing works will be thoroughly included in the next version. Regarding the current version of the paper, I remain reserved in my assessment.

---

> ### Author Response · Authors · 2025-08-05
>
> Thanks for your response!
>
> We wanted to add a few more points about the related works that we didn’t manage to fit in our first response due to space constraints.
>
> **Neural Algorithm Reasoning/CLRS**
>
> Although the works on neural algorithmic reasoning/CLRS share a similar spirit to ours in studying algorithms there isn’t a direct experimental comparison given the differing motivations of these works and ours. While we included [6] in our related works section, we agree that it would be useful to expand the section to include the other CLRS references as well as a more detailed discussion on the differences.
>
> **Neural Sketching**
>
> We’d also like to highlight another crucial difference between our frequency estimation experiments and the neural sketching works [3-5]. We noticed that they add several significant inductive biases to their models to steer them to learn algorithms more similar to traditional sketches. For instance, they parameterize the memory as a 2D grid (M x N dimensional) whereas we use a 1D (K dimensional) grid. Even if we control for size, i.e., set K = MxN, in the multi-query setting, our model still needs to learn to treat its 1D memory as a 2D grid (if that’s optimal) by storing an element in multiple separate positions whereas their setup does that by default. As the authors note, the 2D grid topology is motivated by traditional sketching algorithms such as CountMinSketch [1]. Additionally, when reading from their addressable memory, they also compute the minimum entries of the stored values in order to derive better estimates, a heuristic also derived from the CountMinSketch algorithm. In fact in [5], the authors also combine their learned embeddings with random independent hash functions, one of the fundamental innovations behind the success of sketching algorithms [1,2].
>
> To be clear, we do not view the additional inductive biases as criticisms of their work but rather they highlight a notable difference between our motivation and theirs. Whereas this line of work is focussed on building state of the art sketching methods and thus encouraged to use performance-enhancing optimizations we’re interested in general algorithm discovery from scratch. We intentionally avoid adding additional structure in order to understand what types of algorithms the model can discover by itself. This way, we can simulate whether or not models can find useful algorithms when there is little problem-specific expert knowledge available.
>
>
> [1] Cormode, G. and Muthukrishnan, S. An improved data stream summary: the count-min sketch and its applications. J. Algorithms, 55(1):58–75, 2005.
>
> [2] Charikar, M., Chen, K., & Farach-Colton, M. (2002, June). Finding frequent items in data streams. In International Colloquium on Automata, Languages, and Programming (pp. 693-703). Berlin, Heidelberg: Springer Berlin Heidelberg.
>
> [3] Meta-sketch: A Neural Data Structure for Estimating Item Frequencies of Data Streams. AAAI
>
> [4] Learning to Sketch: A Neural Approach to Item Frequency Estimation in Streaming Data. TPAMI
>
> [5] Lego Sketch: A Scalable Memory-augmented Neural Network for Sketching Data Streams. ICML
>
> [6] Veličković, P., Ying, R., Padovano, M., Hadsell, R., & Blundell, C. (2019). Neural execution of graph algorithms. arXiv preprint arXiv:1910.10593.
>
> In summary, while there are conceptual similarities between these works and ours there are important  differences as well. Ultimately, given the different aims of these works and ours there aren’t clear experimental comparisons to be drawn. Nevertheless, we definitely agree that it's important to include the previous discussions on these works in the next draft in order to better situate our work among existing work. Unfortunately, due to OpenReview restrictions we can’t currently edit the draft, but if accepted, we will definitely do so for the camera-ready version!
>
> We hope our previous response and this one addresses your concerns and clarifies our motivation with this work.
>
> Finally, we wanted to ask if you have any other outstanding concerns so that we can address them during the rebuttal period. Thank you very much!

---

> > ### Author Response · Authors · 2025-08-07
> >
> > Hi,
> >
> > We just wanted to check-In as the rebuttal period ends tomorrow. We hope our above responses clarify the motivation around our work as well as the differences between our work and these related works. Please let us know if you have outstanding concerns that we can address.
> >
> > Thank you again for your consideration!

---

> > > ### Comment · Reviewer_LJYb · 2025-08-08
> > >
> > > Thank you for your response. I believe that the absence of two major lines of related work — including the discussions in your rebuttal and the promised follow-up comparative experiments — poses a significant challenge to both the completeness and the novelty of the paper. I do not believe this can be adequately addressed within the scope of the camera-ready version. Unfortunately, the other reviewers seem to have overlooked this issue. Therefore, I will maintain my score.

---

> ### Author Response · Authors · 2025-08-08
>
> Hi,
>
> We feel there may be a misunderstanding here. As we’ve tried to emphasize in our previous responses, these works are quite different from ours in terms of motivation and scope. We do not believe that either of them compromise the completeness of our experiments and given their substantive differences, the novelty either. We’ve restated the main differences below, please refer to our earlier responses for a more detailed comparison.
>
> As a reminder, our goal with this work is to understand to what extent neural networks can discover data structures from scratch.
>
> ### CLRS
>
> In this line of work, there is no notion of learning data structures. They train networks to simulate specific algorithms which is quite different from our direction of data structure discovery where we're not trying to simulate a specific algorithm. Rather, we want our models to discover data structures by learning algorithms that satisfy certain conditions (i.e. space and query complexity constraints).
>
> There is no clear experimental comparison to be drawn here. It’s unclear how any of these works could be applied to the nearest neighbor and frequency estimation data structure experiments from our work.
>
> The main similarity to our work and this line of work is that we both discuss neural networks and algorithms. As we outline in our related works section, there is in fact a much broader spectrum of works that also explore connections between the two. In our view, this line of work is no different from these other ones. We already referenced one of these works in the CLRS direction [2] and are committed to citing and discussing these other works in the next draft as well.
>
> ### Neural Sketching
>
> Regarding the connection to the neural sketching works, though both our work and theirs include experiments on end-to-end learning of frequency estimation algorithms, there are several crucial differences.
>
> 1. Their emphasis is on building a scalable sketching system whereas ours is on discovering sketching algorithms from scratch. Consequently, they build in several inductive biases from known sketching algorithms in order to facilitate this task (see our previous response for exact details), effectively augmenting a known algorithm (Count-Min Sketch [1]) with learning. By contrast, we intentionally do not do this because we're asking a very different question: can neural networks discover sketching algorithms from scratch?
>
> 2. We approach this from a classical algorithms perspective placing emphasis on theoretical notions of efficiency such as query complexity whereas they are concerned with more practical measures such as throughput.
>
> 3. Finally, frequency estimation is only a small portion of our experiments with a primary focus on nearest-neighbors search. We include it to demonstrate the broader applicability of the framework rather than to propose a SOTA method. Our goal was to understand how to frame data structure learning in a relatively general way and then we derived instantiations of the framework for nearest neighbor search and frequency estimation. From their line of work alone, it’s not clear how to derive this more general paradigm and how to transfer their methods to nearest neighbors.
>
> Regarding an experimental comparison, there may have been a misunderstanding. We did not promise to include these experiments; rather, as noted earlier, we stated that for the next draft we would “either provide a comparison or acknowledge that their method likely performs similarly or better” for frequency estimation. After careful consideration of points (1) and (3) above, we determined that running these experiments is not necessary for evaluating our main claims. Their method does not aim to discover a data structure from scratch, but instead incorporates substantial inductive biases from existing sketching algorithms, making it an unsuitable baseline for evaluating data structure discovery. Moreover, our frequency estimation experiments are not intended to present the best possible sketching method, but to investigate what can be discovered with minimal inductive biases and to illustrate the broader applicability of our framework.
>
>
> Again, we’re still committed to referencing and discussing these works in the next draft and will acknowledge that they are a more scalable solution for frequency estimation.
>
> We hope this response has clearly outlined the substantive differences between these works and ours. While there are some similarities, we strongly do not believe either of them compromise any of the main findings or experiments in our paper. Thank you again for engaging in this review and if you have any questions, please let us know!
>
> [1] Cormode, G. and Muthukrishnan, S. An improved data stream summary: the count-min sketch and its applications. J. Algorithms, 55(1):58–75, 2005.
>
> [2] Veličković, P., Ying, R., Padovano, M., Hadsell, R., & Blundell, C. (2019). Neural execution of graph algorithms. arXiv preprint arXiv:1910.10593.

---

### Official Review · Reviewer_aUtU · 2025-07-02

**Clarity:** 3
**Significance:** 3
**Originality:** 4
**Rating:** 5
**Confidence:** 4

**Summary:**

This paper is primarily about learning data structures for fast nearest neighbor search.

Two models are learned:

1. A transformer that takes as input a data set (consisting of N vectors, each d-dimensional) and outputs a single scalar for each data point. The points are then sorted by this scalar value.

In other words, the transformer selects an *ordering* of the points. The is the *data structure*.

2. A collection of M multilayer perceptrons that take a query point and select possible candidates from the ordered collection of points. This generates a set of M candidates, and of these the nearest (in Euclidean distance) is returned.

To be precise, the jth MLP takes as input the query point as well as the results of the first j-1 MLPs; it then outputs a position (1 to N) in the ordering, identifying the data point at this position.

Various experiments are conducted:

(a) One dimensional data that is uniformly distributed over [-1,1].

The authors take N=100 and M=7. They find that the data structure (approximately) sorts the points in increasing order. Meanwhile the query component does better than binary search: given a query point q, it can approximately predict its position in the ordering, and uses this in the search.

(b) One dimensional data with a harder distribution.

Once again, the data structure ends up sorting the data, but this time the query component uses something like binary search.

(c) Two dimensional data distributed uniformly over [-1,1]^2.

This time, the data structure produces an ordering of the points that is approximately locality-preserving. The authors find it similar to a k-d tree.

(d) Two dimensional data from a harder distribution.

Again, the authors describe the data structure as akin to a k-d tree.

(e) High-dimensional data, e.g. Fashion MNIST.

Here the authors relate the learned data structures to locality-sensitive hashing.

Finally, the authors discuss other applications of the methodology, including a detailed application to frequency estimation in data streams.

**Questions:**

1. In the 1-d case with the uniform distribution:

(a) The paper says that the data structure is 99.5% accurate in perfectly ordering the data. Since N=100, this means that about half the time, the data is perfectly ordered, and half the time, there is one point out of place? Is that correct? How far out of place is that one point?

(b) It would be nice to see in more detail what the query engine is doing. Does it map query q to position (q+1)/2 * N, and then do something like a binary search around that location? I think it would be helpful to see a figure in which, for a particular query (e.g. q = 0.7), the positions of the M probes are shown. This would give a better sense of the querying strategy. In general, the paper does a reasonable job of analyzing the data structures, but not so much the querying strategies.

(c) In this case, although the data structure needs to be relearned for each new data set (from this distribution), it should be possible to keep the querying module fixed... is that right?

2. In the 2-d case with the uniform distribution:

The data structure is some sort of space-filling curve? It would be interesting to see at least a little bit of this curve. The pictures in Fig 9 are great, but the color coding makes it hard to see detail (i.e. there aren't 100 different colors, so the exact ordering in unclear).

For the querying strategy, I really like Fig 10. In fact, Figs 9 and 10 are the most interesting things in the paper and so it is a shame they are in the appendix!

**Ethical Concerns:**

["NO or VERY MINOR ethics concerns only"]

**Final Justification:**

The authors have answered my questions very thoroughly. I maintain my original positive score. I find this paper to be creative, carefully crafted, and insightful. In particular:

* The experimental design makes it possible to carefully study the (neural) data structure and query strategy, and the authors do a good job of this in the 1-d and 2-d settings.
* The learned strategies are fascinating hodgepodges of many data structures that have been developed for NN search over the decades.
* The strategies are a reminder that even in the simplest of cases (e.g. using binary search to identify an element in a sorted list), there are significant gains in efficiency possible from taking the data distribution into account (e.g. permitting a preliminary "guess" of where the item might be).

**Quality:**

4

**Strengths And Weaknesses:**

The good:

1. The overall question (of learning data structure and query engine end-to-end) is fascinating.

2. The choice of nearest neighbor is perfect, as this is a problem with a long history of clever-yet-simple data structures.

3. The choice of data structure -- namely, a permutation of the data -- is sensible.

4. The analysis of the data structures in the 1-d and 2-d setting is quite careful and therefore interesting. In particular, the pictures of the 2-d ordering (in the appendix) are enlightening.

The not-so-good:

1. The 1-d and 2-d data structures and query engines should have been examined in even more detail (see questions later).

2. The higher-dimensional examples have too little detail to be of interest.

3. Likewise with the frequency estimation and other stuff; I think it would be have preferable to skip all of this and just spend more time on the nearest neighbor case.

---

> ### Author Rebuttal · Authors · 2025-07-30
>
> We thank the reviewer for your feedback and appreciate the time and effort you’ve dedicated to evaluating our work! We’re glad to hear you found the direction fascinating and enjoyed the visualizations of the data and query processing! Please find our responses to your questions below. Unfortunately, as we can’t upload a new draft or share any links we cannot provide figures for some of your questions. However, we provide our findings below and will definitely update the next draft to include the corresponding figures.
>
> >(a) The paper says that the data structure is 99.5% accurate in perfectly ordering the data. Since N=100, this means that about half the time, the data is perfectly ordered, and half the time, there is one point out of place? Is that correct? How far out of place is that one point?
>
> Yes, your understanding is correct! We did some additional analysis and found that when the model does make a sorting error, the two points that need to be swapped are neighbors.
>
> >(b) It would be nice to see in more detail what the query engine is doing. Does it map query q to position (q+1)/2 * N, and then do something like a binary search around that location? I think it would be helpful to see a figure in which, for a particular query (e.g. q = 0.7), the positions of the M probes are shown. This would give a better sense of the querying strategy. In general, the paper does a reasonable job of analyzing the data structures, but not so much the querying strategies.
>
> In Figure 2 (center) we show the query model’s behavior on the first lookup. For the 1D uniform distribution, we found that the model starts its search based on its prior (i.e. if q=0.7 and N=100 the model starts at position 70) and then does a local search in that neighborhood in a fashion similar to interpolation search (rather than binary search).
>
> >(c) In this case, although the data structure needs to be relearned for each new data set (from this distribution), it should be possible to keep the querying module fixed... is that right?
>
> Once both networks are trained on datasets/queries from the relevant distributions their parameters no longer need to be updated. After training, given a new dataset (from the same distribution), we simply run a single forward pass through the data-processing network to construct the data structure. The data structure can then be reused multiple times to respond to queries, processed by the query networks. I hope this answers your question, please let us know if you’d like additional clarification!
>
> >The data structure is some sort of space-filling curve? It would be interesting to see at least a little bit of this curve. The pictures in Fig 9 are great, but the color coding makes it hard to see detail (i.e. there aren't 100 different colors, so the exact ordering in unclear).
>
> That’s a great question and yes your intuition is correct! We plotted the data structure model’s decision boundary and we see that it roughly divides the space into evenly sized partitions.
>
> We hope this response answers your questions and addresses your concerns. Please let us know if you have any other questions!

---

> > ### Comment · Reviewer_aUtU · 2025-08-08
> > **Author comments**
> >
> > To the authors: thanks for your comments. I like this paper very much, and have added comments to my review accordingly. To re-emphasize: the most interesting thing here is the insight into the nature of the learned ordering and query strategy. This comes out most clearly in the 1-d and 2-d case, and if it is possible to do more for the higher-dimensional setting, that would be even better.

---

> > > ### Author Response · Authors · 2025-08-08
> > >
> > > Thank you very much for your kind comments, we’re very glad that you enjoyed the paper!  Initially, we weren’t sure if the models would make progress even in the 1d/2d settings and it was interesting trying to understand the algorithms they discovered. We definitely believe there’s more potential for future work to try and reverse-engineer these algorithms!

---

### Official Review · Reviewer_LhkM · 2025-07-23

**Clarity:** 3
**Significance:** 3
**Originality:** 3
**Rating:** 4
**Confidence:** 4

**Summary:**

This paper introduced a method for learning data structures for nearest-neighbor search and count estimation. The data processing and query execution networks are learned in an end-to-end manner. The data is processed by a NanoGPT-based transformer, which returns a score for each data point that can be used to order the points. The model uses a differentiable sort function during training, allowing it to optimize the ordering of the data specifically for nearest-neighbor search.
The query execution network performs queries sequentially, where each model takes the query and the history of previous selections to output a one-hot vector that selects a data point. Given a budget of 𝑀 lookups, the query execution model aims to find the closest data point at each step. The network is trained using softmax-based soft lookups with added noise to encourage sparse selections. During inference, the softmax output is replaced with a hardmax operation to produce the final one-hot encoding for exact lookups.
The authors have discussed 2 use cases with this method:
1. Near neighbor search: For 1 dimensional data, 2 dimensional data and n dimensional data (with different distributions)
2. Frequency estimation

**Questions:**

1. How does the comparison looks like with SOTA NN search methods (graph based search like HNSW)?
2. Does number of lookups translates well to the query latency for the proposed approach? (Is it a linear relationship?)

**Ethical Concerns:**

["NO or VERY MINOR ethics concerns only"]

**Final Justification:**

Issues not resolved (but stated as not in the scope of the paper to claim)
1. real world datasets (from ANN/BigANN benchmarks):
2. RecallK@K measures: not resolved (accuracy is reported),
3. Comparison with SOTA NN search methods (graph based search like HNSW)

Issues resolved :
1. Comparison of data structure sizes
2. Construction (+learning) time and query latency (wall clock time and comparison on number of lookups).
3. Performance metrics with data dimensions.
4. Other potential applications discussion
5. Does number of lookups translates well to the query latency for the proposed approach? (Is it a linear relationship?)

**Limitations:**

Yes

**Paper Formatting Concerns:**

I didn't see any formatting issues in the paper.

**Quality:**

3

**Strengths And Weaknesses:**

Strengths:
1. The paper has introduced a novel idea of learning a data structure via transformers. The learning framework allows the data structure to adapt to the data distribution automatically.
2. The idea is well motivated and analysis is provided on the behavior of data structure becoming similar and yet better than existing data structures like binary search, KD trees and LSH.
3. The model handles the challenge of learning with discrete lookups by using softmax with noise during training and switching to hard softmax at the inference.

Weaknesses:
1. The paper needs more detailed experiments and analysis to validate the novelty of the idea. For example, the near neighbor search analysis will need more real world datasets (from ANN/BigANN benchmarks), RecallK@K measures, comparison of data structure sizes, construction (+learning) time and query latency (wall clock time and comparison on number of lookups).
2. The paper doesn't give a clear picture on how the performance metrics vary with data dimensions.
3. Other potential applications: The discussion in this section is vague. It lacks concrete examples or technical details for how this approach could be adapted to domains beyond nearest-neighbor search, making the section feel speculative and underdeveloped.

---

> ### Author Rebuttal · Authors · 2025-07-30
>
> We thank the reviewer for your feedback and appreciate the time and effort you’ve dedicated to evaluating our work! Before responding to specific points, we’d like to clarify the high-level motivation behind our work.
>
> We definitely understand the concerns around the efficiency and scalability of our methods and acknowledge that it’s necessary for future work to address this in order to build a system suitable for production workloads. Our current methods are not designed to scale to large datasets and we believe that more problem-specific inductive biases are crucial for this (see App D for a discussion).
>
> We’d like to emphasize, however, that **our intention with this work was not to propose a method built for real-world deployment. Instead, we are interested in understanding to what extent neural networks can explore the space of data structures—even at small scale—from a classical algorithms perspective.**
>
> Initially, it was unclear (at least to us) whether end-to-end learning of data structures for tasks like nearest neighbors was even feasible, or how to even frame the problem. It raises many fundamental questions: How should data structures and queries be represented using neural networks, especially under combinatorial constraints like limited query budgets? Will training succeed at all, given that the data-processing and query networks are trained from scratch and rely entirely on each other? Even then, can such a system explore the space of classical and novel algorithms?
>
> To our surprise, once we carefully set up training—with space and query constraints and a structured lookup interface—the model was able to discover meaningful algorithms! It recovered classic data structures like sorting, KD-trees, and LSH-like behavior, often adapting them to the underlying distribution.
>
> To better understand what the model is learning, we place a strong emphasis on reverse-engineering its behavior and comparing it to classical algorithms. This perspective also motivates our focus on theoretical notions of efficiency such as query and space complexity—quantities we can directly enforce and optimize during training. In contrast, metrics like wall-clock time are often closely tied to hardware and implementation details and were not the central focus of our investigation.
>
> To summarize, we view this as our core contribution: framing data structure discovery as an end-to-end learning problem and showing that it can explore a rich algorithmic space.  Finally, to our knowledge, this is one of the first attempts to approach data structure discovery in a general way. While our experiments focus on NN search, the design principles extend to other algorithmic settings. Of course, scaling to real-world workloads is an exciting next step, but our results show that even small-scale models can yield useful insights—potentially informing the design of scalable, hand-engineered systems
>
> We hope this clarifies our intentions with this work. Nevertheless, we’ve measured the throughput of our models for the nearest neighbor experiments in several regimes.
>
>
> ### Performance Metrics
> We emphasize that we did not optimize for practical efficiency as we were primarily interested in the fundamental tradeoffs between space and query complexity, though these are certainly related to runtime in practice. We use the same architectures from our paper (see App B.1 for details). Benchmarking is conducted with a batch size of 512 and the results are averaged over 10 trials on a single GPU.
>
> **Query Processing Throughput**
>
> Number of queries processed per second when using log(n) lookups per query. As expected, querying smaller datasets has higher throughput as the number of lookups, and thus neural net forward inferences, is smaller.
>
> | dim  | n=64   | n=128  | n=256  | n=1024 | n=2048 |
> |------|--------|--------|--------|--------|--------|
> | 32   | 212943 | 175559 | 146495 | 113960 | 103476 |
> | 64   | 211518 | 173807 | 149603 | 113249 | 105833 |
> | 128  | 208741 | 172275 | 151533 | 116607 | 104858 |
>
> Below, we fix N=256 and measure how increasing the number of lookups (M) affects throughput. There’s a clear inverse correlation between M and throughput indicating that query complexity has implications for practical efficiency.
>
> | dim  | M=1     | M=2    | M=3    | M=4    | M=5    | M=6    | M=7    | M=8    |
> |------|---------|--------|--------|--------|--------|--------|--------|--------|
> | 32   | 1072476  | 517067 | 345946 | 261705 | 212202 | 176503 | 145927 | 134165 |
> | 64   | 1090290 | 513129 | 355654 | 265753 | 211991 | 174031 | 150917 | 132594 |
> | 128  | 1054366 | 519691 | 348347 | 260985 | 212237 | 173406 | 145191 | 133278 |
>
> **Data-preprocessing Throughput**
>
> The table below shows the number of raw datasets processed into a data structure per second using quadratic and linear attention (which we explored as a faster alternative, see App. D.1).  Quadratic attention runs faster here due to flash-attention [7], which optimizes memory access. Notably, in smaller N regimes, linear attention can be less efficient than quadratic attention with hardware optimizations, despite its lower FLOP count [7].
>
> | Arch         | Dim | n=64  | n=128 | n=256 | n=1024 | n=2048 |
> |--------------|-----|-------|-------|-------|--------|--------|
> | **Linear**   | 32  | 17701 | 8772  | 4542  | 1170   | 585    |
> | Linear       | 64  | 17241 | 8844  | 4563  | 1177   | 590    |
> | Linear       | 128 | 17121 | 8614  | 4473  | 1152   | 577    |
> | **Quadratic**| 32  | 95229 | 49697 | 13418 | 5867   | 571    |
> | Quadratic    | 64  | 94815 | 46924 | 13285 | 5248   | 548    |
> | Quadratic    | 128 | 93294 | 48506 | 13095 | 5174   | 543    |
>
> We also try fixing the input dimension to 128 and adjust the number of layers in the quadratic transformer.
>
> | # Layer | n=64   | n=128  | n=256  | n=1024 | n=2048 |
> |---------|--------|--------|--------|--------|--------|
> | 4       | 174433  | 174381 | 173501 | 31512  | 8449   |
> | 8       | 94493  | 95037  | 18849  | 6780   | 610    |
> | 12      | 65331  | 24301  | 8073   | 1103   | 220    |
>
> **Training Time**
>
> Below is the training time for each of our experiments with N=100 sized datasets
> | 1D        | 2D        | 30D        | MNIST      | SIFT       | FashionMNIST |
> |-----------|-----------|------------|------------|------------|--------------|
> | ~3 Hours  | ~3 Hours  | ~10 Hours  | ~10 Hours  | ~2 Hours   | ~2 Hours     |
>
>
> >How does the comparison looks like with SOTA NN search methods (graph based search like HNSW)?
>
> As discussed in the overall motivation for our work above, we do not include comparisons to SOTA NN methods such as HNSW as these are complex systems explicitly designed to maximize runtime efficiency in practice which is not the intention of our work.
>
> >Does number of lookups translates well to the query latency for the proposed approach? (Is it a linear relationship?)
>
> Yes! The number of lookups is inversely proportional to query throughput (see tables provided above). Although the relationship is not exactly linear in our measurements this is likely due to some constant overhead as the relationship should asymptotically be linear. This is because each additional lookup requires an additional forward pass through a query model.
>
>
> ### Other Potential Applications
> >Other potential applications: The discussion in this section is vague. It lacks concrete examples or technical details for how this approach could be adapted to domains beyond nearest-neighbor search, making the section feel speculative and underdeveloped.
>
> Thanks for your feedback. Our goal in the “Other Potential Applications” section is simply to show that a wide range of problems—beyond nearest‑neighbor search—can be framed as “data‑structure discovery” and tackled with similar methods to those which we use.  We do not go into too much depth as a full treatment of each would indeed deserve its own paper. However, we kindly ask the reviewer to refer to appendix E where we have provided more detail on how the framework can be applied to these other problems including technical details such as architecture choices. As an example, we can describe how to adapt the framework to sparse matrix problems. Here, the goal is to efficiently answer queries about an NxN matrix (e.g., a matrix-vector product). The data network compresses the matrix into K tokens (with K < N); the query model retrieves M data chunks based on a query vector, and the predictor network computes the final response (e.g., learning to perform the product). The same architectures from NN search and frequency estimation can be used for this problem. Please let us know if you’d like additional clarification.
>
>
> We hope this response answers your questions and addresses your concerns. Please let us know if you have any other questions!

---

> > ### Author Response · Authors · 2025-08-05
> >
> > Hi,
> >
> > As the rebuttal period is ending soon, we just wanted to follow-up and check if our response addresses your concerns and clarifies the motivation around our work. Please let us know if we can provide any additional clarification or if you have outstanding concerns.
> >
> > Thanks for your time!

---

> > > ### Comment · Reviewer_LhkM · 2025-08-06
> > >
> > > Thanks to the authors for addressing my questions.
> > > All my concerns except the benchmarking against performant NN systems and with larger datasets are addressed.
> > > But, I understand author's comment on the motivation behind the work, and hence increasing my score.

---

> > > > ### Author Response · Authors · 2025-08-08
> > > >
> > > > Thanks very much for your reconsideration! We’re glad to hear we were able to address several of your concerns and clarify our motivation. We believe that scaling these systems is an interesting direction for future work!

---

### Note · Authors · 2025-08-12

Dear Reviewers,

First, we would like to sincerely thank you for your careful reading of our submission and for the thoughtful feedback you have provided throughout the process.

We were glad that over the rebuttal period we were able to clarify our motivation with this work as trying to understand to what extent neural networks can explore the space of data structures from scratch.

We wanted to briefly discuss a point from our exchange with one of the reviewers. In their comments, the reviewer expressed concern that our work is missing experimental comparisons to two lines of related research—CLRS and neural sketching. We believe these works are fundamentally different in scope and goals from ours, and therefore not suitable comparisons for our main claims.

During the rebuttal period, we provided several detailed clarifications explaining these differences and committed to citing both lines of work more thoroughly in the next draft. In short:

- **CLRS** is only loosely related in that it also involves algorithms and neural networks, but it does not have practical implications for our experiments.


- **Neural sketching** has very different motivations and overlaps only with a small subset of our results.

We have outlined these distinctions in detail in our forum comment [here](https://openreview.net/forum?id=DOaqwuhEjc&noteId=Kcs2lvqIlB), as well as in earlier responses to the reviewer. We believe that neither line of work undermines the novelty, experimental validity, or main findings of our paper.

We wanted to ensure you had this context when evaluating both the review and our responses. Thank you again for your time and for your efforts in the reviewing process!

---

### Decision · Program_Chairs · 2025-09-17

**Decision:**

Accept (poster)

**Comment:**

This paper proposes a method to learn data structures end-to-end with neural networks, focusing on nearest-neighbor (NN) search but also including frequency estimation. It has received mostly positive reviews.

The paper was found clearly organized, well written, fluent, easy to read, with clear figures and tables. The problem was found challenging, fascinating, with high impact. The choice of NN search was found perfect. The method was found well motivated, novel, interesting, creative, sensible, simple, effective, technically solid. The analysis was found interesting, insightful. The experiments were found sufficient, with thorough ablations. The performance was found competitive with classical data structures. The findings were found very interesting

Concerns included missing related work, relying on large architectures incurring significant cost, scalability, insufficiency of large datasets, metrics and ablations, data dimensions being small, the section on other applications being vague and underdeveloped, lack of detail, the frequency estimation part unclear and recommended to skip, absence of worst-case guarantees

The authors provided detailed feedback in their rebuttal, accompanied with new experiments where needed. This feedback addressed most of reviewers' concerns and substantial discussion followed. The authors committed to updating the paper with the new material they prepared during rebuttal and discussion.

Concerns that remain after discussion include insufficiency of large datasets, lack of RecallK@K measures, comparison with SOTA NN search methods (Reviewer LhkM) and missing related work CLRS and Neural Sketches (Reviewer LJYb, who remained negative).

On the former, the authors argued that they do not to propose a method built for real-world deployment but rather to investigate to what extent neural networks can explore the space of data structures. The reviewer understood the motivation behind the work and remained positive.

On the latter, very long discussion followed between the authors and the reviewer, as well as the reviewer and myself. The relation to CLRS appears to have been addressed. However, the reviewer found that the relation to Neural Sketches, a recent work, is critical.

The authors accepted that there are similarities between their work and Neural Sketches, but also differences, such that the novelty of this work is not challenged. Their work focuses mainly on NN search while Neural Sketches are specifically designed for frequency estimation. They have committed to discuss Neural Sketches (as well as CRLS) and to "acknowledge that they [Neural Sketches] are a more scalable solution for frequency estimation". The reviewer has insisted that "the authors have overlooked the broader field of memory-augmented neural networks" and suggested that the author adapt Neural Sketches to NN search, which however does not appear to be straightforward or within the scope of this work.

Given the very positive position of the other reviewers and the investigative nature of this work on a challenging problem that has not been addressed before, I find that this work has merit for publication and I recommend that it is accepted. However, given the discussions, I recommend that the authors move "other applications" discussions (section 3.2) to the supplementary and then either add detailed design and comparison with Neural Sketches if possible in the frequency estimation part (section 3.1) or entirely move the frequency estimation part to the supplementary as well, leaving space more space for the new material and strengthening the NN part.